# mTOR controls ependymal cell differentiation by targeting the alternative cell cycle and centrosomal proteins

Alexia Bankolé[1,7], Ayush Srivastava[2,7], Asm Shihavuddin [3,4], Khaled Tighanimine [1], Marion Faucourt[2], Vonda Koka[1], Solene Weill[2], Ivan Nemazanyy[5], Alissa J Nelson[6], Matthew P Stokes[6], Nathalie Delgehyr[2], Auguste Genovesio[3], Alice Meunier[2], Stefano Fumagalli [1], Mario Pende [1✉] & Nathalie Spassky [2✉]

## Abstract

**Ependymal cells are multiciliated glial cells lining the ventricles of the mammalian brain. Their differentiation from progenitor cells involves cell enlargement and progresses through centriole amplification phases and ciliogenesis. These phases are accompanied by the sharp up-regulation of mTOR Complex 1 activity (mTORC1), a master regulator of macromolecule biosynthesis and cell growth, whose function in ependymal cell differentiation is unknown. We demonstrate that mTORC1 inhibition by rapamycin preserves the progenitor pool by reinforcing quiescence and preventing alternative cell cycle progression for centriole amplification. Overexpressing E2F4 and MCIDAS circumvents mTORC1-regulated processes, enabling centriole amplification despite rapamycin, and enhancing mTORC1 activity through positive feedback. Acute rapamycin treatment in multicentriolar cells during the late phases of differentiation causes centriole regrouping, indicating a direct role of mTORC1 in centriole dynamics. By phosphoproteomic and phosphomutant analysis, we reveal that the mTORC1-mediated phosphorylation of GAS2L1, a centrosomal protein that links actin and microtubule cytoskeletons, participates in centriole disengagement. This multilayered and sequential control of ependymal development by mTORC1, from the progenitor pool to centriolar function, has implications for pathophysiological conditions like aging and hydrocephalus-prone genetic diseases.**

**Keywords** mTOR; Ciliogenesis; Differentiation; Cell Cycle; Cytoskeleton
**Subject Categories** Cell Adhesion, Polarity & Cytoskeleton; Development

## Introduction

Multiciliated ependymal cells (MCCs) are glial cells that line all the ventricular cavities of the vertebrate brain. They have dozens of motile cilia on their surfaces, which beat synchronously to contribute to the flow of cerebrospinal fluid (CSF) through the cavities and thus maintain the homeostasis of the ventricular system. Locally, these ciliary beats eliminate toxic waste and debris and ensure the correct distribution of metabolites and signaling molecules to neighboring cells. At the tissue level, the ependymal epithelium forms a permeable barrier to ions, water, and immune cells between the ventricle and the cerebral parenchyma. Along the lateral ventricles, post-mitotic ependymal cells contribute to adult neurogenesis by regulating the proliferation of neural stem cells (NSCs) (Obernier and Alvarez-Buylla, 2019). Thus, ependymal cells play significant roles during brain development and in the adult. Accordingly, several neurological disorders are linked to ependymal cell impairment (Cruchaga et al, 2013; Keryer et al, 2011; Lambert et al, 2013; Silverberg et al, 2003). Understanding how these cells develop will be useful in elucidating the etiology of these disorders. In particular, how cell proliferation, size, shape, and differentiation are coordinated remains undeciphered.

Ependymal and adult NSCs derive from radial glial cells (RGCs) at late embryonic stages, through either consuming divisions to generate two ependymal cells, or asymmetric divisions, giving rise to one ependymal cell and one adult NSC. This glial fate choice of RGCs is governed by the Geminin family proteins (GEMC1, MCIDAS, and GEMININ) (Ortiz-Alvarez et al, 2019). GEMININ, a well-known cell cycle regulator (Wohlschlegel et al, 2000) binds to Cdt1 and prevents DNA re-replication, favoring the adult NSC fate. In contrast, GEMC1 and MCIDAS, in association with E2F4/5 and DP1, are essential for initiating and progressing the multiciliation program (Kyrousi et al, 2015; Lewis and Stracker, 2021; Vladar et al, 2018). GEMC1, the most upstream regulator of multiciliogenesis induces DNA damage during the G1/S phase in RGCs, activating the p53-p21/p73 signaling pathway before centriole

[1]Université Paris Cité, CNRS, INSERM, Institut Necker Enfants Malades-INEM, F-75015 Paris, France. [2]Institut de Biologie de l'Ecole Normale Superieure (IBENS), Ecole Normale Supérieure, CNRS, INSERM, Université PSL, Team Cilia Biology and neurogenesis, 75005 Paris, France. [3]Institut de Biologie de l'Ecole Normale Superieure (IBENS), Ecole Normale Supérieure, CNRS, INSERM, Université PSL, Team Computational bioimaging and bioinformatics, 75005 Paris, France. [4]Department of EEE, Presidency University, Dhaka, Bangladesh. [5]Platform for Metabolic Analyses, Structure Fédérative de Recherche Necker, INSERM US24/CNRS UAR 3633, Paris, France. [6]Cell Signaling Technology INC, 3 Trask Lane, Danvers, MA 01923, USA. [7]These authors contributed equally: Alexia Bankolé, Ayush Srivastava. ✉E-mail: mario.pende@inserm.fr; nathalie.spassky@bio.ens.psl.eu

amplification (Ortiz-Alvarez et al, 2022). Subsequently, nuclear deformations trigger G1/S transition through RB1 phosphorylation and MCIDAS activation (Basso et al, 2024). Recently, a cell cycle variant has been uncovered in MCC progenitors, where the sequential expression of G1/S/G2/M phase proteins, notably cyclin-dependent kinase 2 (Vladar et al, 2018) and 1 (Al Jord et al, 2017), along with most of their cognate cyclins, sustains centriole amplification. Strikingly, the temporal expression of cyclins E2 and A2 are replaced by non-canonical cyclins O and A1 and E2F7 transcription factor, to secure DNA replication block during MCC differentiation (Choksi et al, 2024; Khoury Damaa et al, 2025; Serizay et al, 2025). In the airways, centriole amplification is accompanied by apical surface expansion, which is scaled by the number of centrioles that occupy the entire apical surface of the cells (LoMastro et al, 2022; Nanjundappa et al, 2019). In contrast, the centriolar patch occupies only one-third of the apical surface of mature ependymal cells (Mirzadeh et al, 2010). Although the apical surface of ependymal cells increases during their differentiation, the mechanisms governing cell size during development are still unknown.

The mammalian Target of Rapamycin (mTOR) kinase is a master regulator of growth and metabolism, integrating growth factors and environmental nutrient signals (Liu and Sabatini, 2020). mTOR exists in the cell in at least two distinct protein complexes, mTORC1 and mTORC2, with different sensitivities to upstream signals, intracellular localization, and protein substrates. Both mTORC1 and mTORC2 are activated downstream of tyrosine kinase growth factor receptors, but mTORC1 is also selectively activated by nutrient levels sensed at the external lysosomal membranes. The allosteric mTOR inhibitor rapamycin is a natural macrolide that preferentially affects specific mTORC1 targets (Thoreen et al, 2009; Yip et al, 2010). The ubiquitous expression of mTOR in eukaryotic cells underlies its essential function of promoting macromolecule biosynthesis, depending on nutrient availability. Thus, mTOR activation in proliferating and post-mitotic cells increases biomass accumulation and cell size. In cycling cells, mTOR has been shown to cross-talk with the cell cycle machinery and regulate the progression through different cell cycle phases (Joshi et al, 2024; Rubin et al, 2020). mTOR is also present in RGCs and increases the apical contact size during corticogenesis (Foerster et al, 2017). Thus, addressing whether mTOR impacts cell size and the alternative cell cycle during multiciliated cell differentiation is essential.

Aberrant activation of mTOR underlies mTORopathies and brain malformations characterized by abnormal cell architecture and morphology, leading to megalencephaly, epilepsy, autism, and other comorbidities (Hasbani and Crino, 2018). Somatic and mosaic mutations in Akt3, PIK3CA, mTOR, TSC1, TSC2, DEPDC5, and Rheb genes underlie these genetic diseases. While Akt3, PIK3CA, and mTOR gain-of-function mutations result in hyperactivity of both mTORC1 and mTORC2, mutations in TSC1, TSC2, DEPDC5, and Rheb lead to the selective activation of mTORC1. Thus, mTORC1 hyperactivity is sufficient to trigger brain abnormalities. A significant source of TSC-related morbidities are periventricular lesions that arise from progenitors during gestation and the first years of life. Their abnormal growth may lead to hydrocephalus, a hallmark of ependymal impairment in mice (Zhang et al, 2006), increased intracranial pressure, and death.

Thus, identifying the mTORC1 targets that regulate ependymal cell differentiation may open new avenues of intervention.

In this study, we modulate the mTORC1 pathway by pharmacological and genetic interventions in vivo and in vitro to investigate the impact on the ependymal cell cycle variant and differentiation. We show that mTORC1 inhibition by rapamycin promotes a quiescent state, dampening the expression of cell cycle proteins from G1 through G2 phases. Consequently, most cells failed to differentiate, while the others had a smaller apical surface and fewer centrioles. Increasing mTOR activity leads to the opposite phenotypes. We also address the epistatic interaction with master regulators of ependymal cell differentiation (MCIDAS/ E2F4) and with one of the downstream effectors of mTOR1 (S6 Kinase). To understand the consequences of mTOR inactivation, we analyze the metabolome and the phosphoproteome during ependymal cell differentiation in the presence or absence of Rapamycin. We reveal multiple targets of mTORC1 with enrichment in proteins involved in the cell cycle and cytoskeletal dynamics. In particular, we provide evidence that the actin and microtubule cytolinker GAS2L1 protein controls centriolar number and density downstream of the mTORC1 pathway.

# Results

## Morphological changes during ependymal cell differentiation

Ependymal cells follow a consistent spatiotemporal differentiation pattern on the lateral walls of the lateral ventricles of the mouse brain. Starting at P0, differentiation begins at the caudal end and the ventral border of the wall, which then spreads along a caudo-rostral and ventro-dorsal gradient (Spassky et al, 2005). Brains were examined to investigate the morphological dynamics of the lateral wall during ependymal cell differentiation. Cen2GFP (centriolar marker) lateral ventricular walls at various postnatal ages were immunostained with Sas6 to label centrioles in formation and β-catenin for cellular junctions (Fig. 1A). All the cells on the ventricular wall were segmented based on β-catenin staining and classified into the corresponding stages of differentiation as previously described (Shihavuddin et al, 2017; Spassky and Meunier, 2017). Briefly, cells with two dots of Cen2GFP are progenitor cells; the presence of Sas6 and a cloud (Amplification phase) or ring-like structures (Growth phase) of Cen2GFP indicates cells in the process of differentiation; multiple dots of Cen2GFP indicate cells that have completed the centriole amplification with their centrioles disengaging from each other (Disengagement phase) or docked at the plasma membrane with motile cilia (Ependymal cells) (Fig. 1B). A similar differentiation path is apparent when staining centrioles and deuterosomes with anti-FOP and -Deup1 antibodies, respectively (Fig. 1C). As previously described, the overall wall size increases with age despite a decrease in the number of cells contacting the ventricle (Fig. 1D,E) (Redmond et al, 2019). This is explained by ependymal cells expanding their apical surface with differentiation (Fig. 1F) and increasing proportion of fully differentiated ependymal cells with age (Fig. 1G). Thus, the expansion of the apical contact of the cell is a cue of ependymal development.

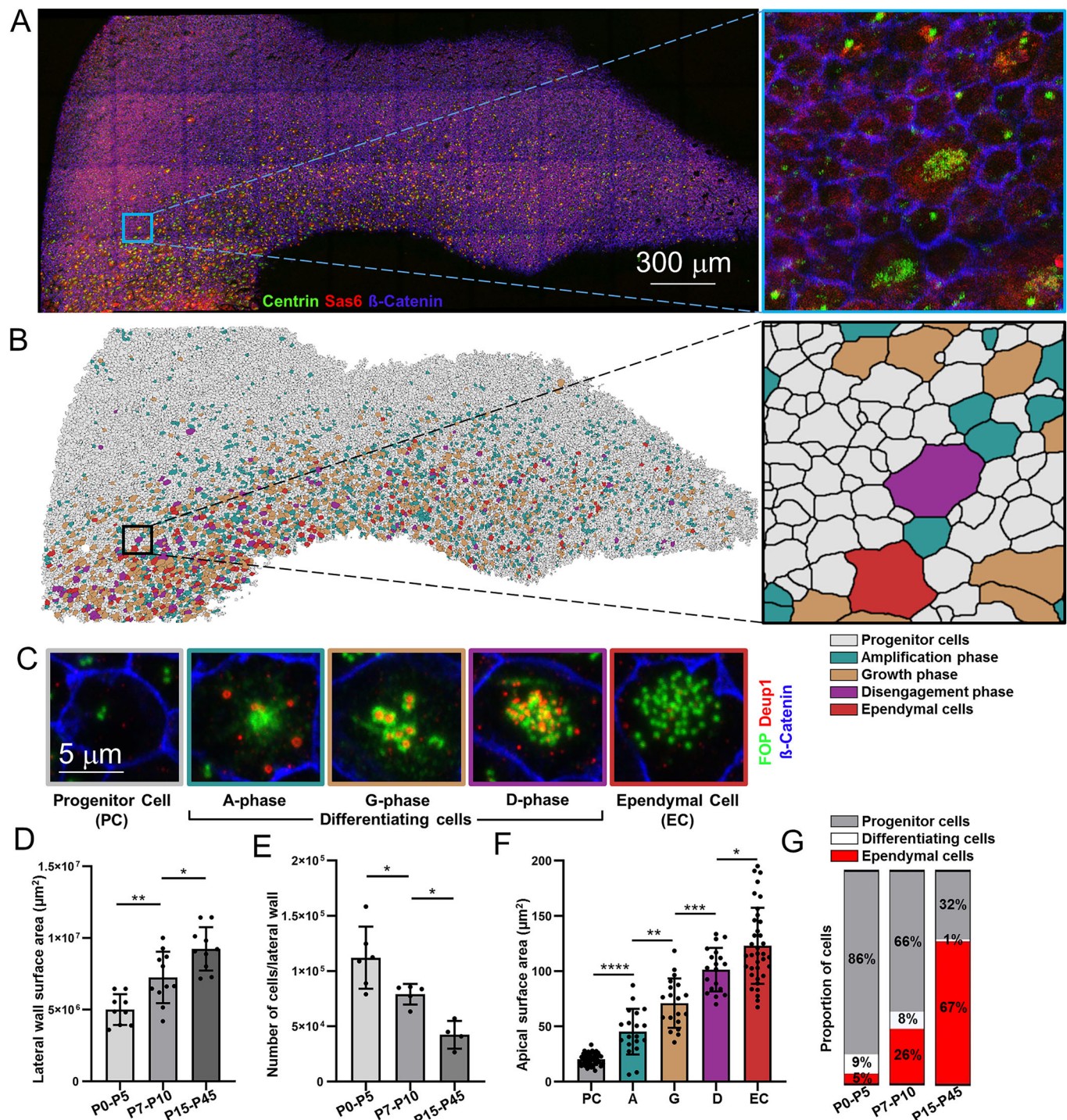

**mTORC1 activity controls the size of ependymal apical contacts and the number of centrioles**

mTORC1 activation stimulates cell growth in many organs (Liu and Sabatini, 2020), and is activated in differentiating ependymal cells (Kimura et al, 2021). To study the precise stages of mTORC1 activation during centriole amplification, the lateral wall at different postnatal ages was immunostained for p-rpS6 (readout for

mTORC1 activity), cilia, centrioles, and p21 (a cyclin-dependent kinase inhibitor and a marker of differentiating ependymal cells (Ortiz-Alvarez et al, 2022)). For all ages examined, p-rpS6 activation was observed exclusively during centriole amplification stages and was tightly correlated with the expression of p21 (as shown, for example, on postnatal day 4 (P4) lateral wall in Fig. EV1). We then used gain and loss-of-function approaches to study the role of mTORC1 activation during in vivo ependymal cell

**Figure 1. Ependymal cells grow apically with multiciliation.**

(A) The complete lateral ventricular wall is labeled at P1 with SAS6 (deuterosomes), CENTRIN (centrioles), and ß-Catenin (cell junctions). (B) Segmented lateral ventricular wall color-coded as white cells- progenitor cells, turquoise cells- A-phase, fawn cells- G-phase, purple cells- D-phase, red cells- ependymal cells. Insets of 'A' and 'B' show acquired image and segmentation in higher magnification. (C) Cells in their respective phases of differentiation are characterized by their labeling of FOP (centrioles), DEUP1 (deuterosomes), and ß-catenin (cell junctions). (D) The area of the lateral wall is calculated by measuring the size of dissected tissue for each age group (each dot represents one wall of the lateral ventricle, $n > 9$; **$P = 0.0031$; *$P = 0.0197$). (E) The total number of cells on the lateral wall is calculated by extrapolating data from segmented images for each age group (each dot represents one wall of the lateral ventricle, $n > 4$; *$P = 0.03$; *$P = 0.0159$). (F) The apical surface area of cells in different phases of differentiation is calculated by segmentation based on cell junction markers (each dot represents one cell, $n > 3$; ****$P < 0.0001$; ***$P = 0.0001$; **$P = 0.001$; *$P = 0.0266$). (G) The proportion of cells is based on their differentiation status and age. Cells from A-phase, G-phase & D-phase are combined under "differentiating cells". Data information: In (D–F), data are presented as mean ± SD. ****$P < 0.0001$, ***$P < 0.001$, **$P < 0.01$, *$P \leq 0.05$ (Student's $t$ test). Source data are available online for this figure.

differentiation. mTORC1 inhibition through daily subcutaneous injections of rapamycin for 4 days from P0 (differentiation onset at the lateral ventricle) led to a significant decrease in the number and size of ependymal cells, which was counteracted by the increased number of progenitors with a larger apical surface (Fig. 2A–C). This implies that mTORC1 is necessary to kick-start ependymal cell differentiation and growth. As cell growth is concurrent with centriole amplification (LoMastro et al, 2022; Nanjundappa et al, 2019), we analyzed the result of centriolar amplification by counting the number of centrioles at the end of the process, i.e., in mature cells. We found that the number and area occupied by the centrioles decreased with rapamycin treatment (Fig. 2D–F). To study the long-term effects of a 4-day rapamycin treatment, we analyzed ventricular morphogenesis at P10 and P30. At P10, the inhibitory effects of rapamycin on differentiation are still visible, with an increased proportion of immature cells (stem cells and differentiating cells) and a decreased proportion of mature cells (Fig. EV2A,B). This effect on differentiation is no longer visible at P30, indicating a recovery of differentiation at this time point. However, while the cells undergo growth between P4 and P30 (compare Fig. 2C and EV2), the ependymal cell size remains smaller, suggesting that the effects of rapamycin on cell growth do not recover over time. Taking a different approach, we activated mTORC1 by genetically targeting Tsc1, part of the mTORC1 upstream negative regulatory complex TSC (Menon et al, 2014). The genetic mutant, Tsc1$^{\text{lox/ko}}$ Nestin-Cre $^{+/-}$ (Tsc1 cKO), was analyzed only at P0 due to perinatal lethality. Constitutive mTORC1 activation, confirmed by increased p-rpS6 staining in progenitor cells (negative for the FoxJ1 marker of ependymal differentiation) in Tsc1 cKO compared to controls, validated the proof of concept (Fig. 2G,H). The number of centrioles per cell and the area of their patch were increased in Tsc1 cKO at P0 (Fig. 2I–K). Centriole number and multiciliated cell surface area are correlated in the airways (LoMastro et al, 2022; Nanjundappa et al, 2019). We thus studied this relationship in multiciliated ependymal cells among controls at birth, which is weak ($R^2 = 0.34$) and tends to decrease at P4 ($R^2 = 0.15$). However, the slope of the line is statistically significant, indicating that this correlation exists, although other factors may also contribute to regulating ependymal cell size. Interestingly, mTORC1 modulation results in a lower correlation at birth (Tsc1 cKO: $R^2 = 0.19$) or even a complete loss of correlation at P4 (Rapamycin: $R^2 = 0.06$), suggesting that changes in cell size alone cannot explain the phenotypes observed in our study (Fig. EV1). Thus, mTORC1 activation through Tsc1 inactivation or inhibition by a 4-day daily injection of rapamycin can modulate the number of centrioles produced in ependymal cells and the

corresponding size of their apical area. This suggests that mTOR controls the timing of centriole amplification, the increase in apical contact size, and the number of centrioles.

## Ependymal cell differentiation and centriole disengagement are controlled by mTORC1 activity in vitro

We then used ependymal cell cultures to delve further into the mechanisms of mTORC1 regulation of ependymal cell differentiation. Various stages of basal body formation during ependymal cell differentiation were detectable in vitro, mirroring the in vivo analysis. The amplification stage (A-phase) corresponds to the formation of procentrioles around deuterosomes; the growth stage (G-phase) is when procentrioles grow, and the disengagement stage (D-phase) is when centrioles separate and migrate apically to become basal bodies (Fig. EV3A) (Al Jord et al, 2019). Cells were treated with rapamycin at the time of the switch to the differentiation medium (Day 0) and analyzed after 2, 4, or 8 days of treatment. Rapamycin treatment abolished p-rpS6 immunostaining and inhibited ependymal cell differentiation, as observed by the decreased number of FoxJ1 or multicentriolar positive cells (Fig. 3A–D). These cells displayed fewer centrioles in a smaller area (Fig. 3E–G). Time-lapse movies of differentiating cells under acute short rapamycin treatment were made to investigate how rapamycin affects the centrioles. Rapamycin was added to Cen2GFP$^+$ cells at Dif 4, and cells were filmed 2 h later when the p-rpS6 staining was no longer visible. At this point, some cells are in the A- and G-phase, allowing us to assess the effect of rapamycin on differentiating cells. Rapamycin treatment resulted in deuterosomes regrouping and impaired centriole ability to disperse from their platforms (Fig. 3H,I), without affecting the actin and microtubule cytoskeletons (Fig. EV3B). Rapamycin is an allosteric inhibitor of mTORC1, which has differential effects on mTORC1 targets, notably the canonical and non-canonical mTORC1 targets (Napolitano et al, 2020). Next, we assessed a catalytic inhibitor of mTOR, Torin 1, on cultured ependymal cells. Treatment with Torin 1 decreased p-rpS6 expression and the number of multiciliated cells at Dif 4 and Dif 8, reflecting the effects of rapamycin (Fig. EV3C). Thus, canonical mTORC1 inhibition affects several stages of ependymal differentiation, such as the initiation of differentiation, the number of centrioles formed, and their ability to disperse in the cell. At early time points, rapamycin decreases centriole amplification, reducing the number of centrioles. In addition, it opposes centriole dispersion at later time points, resulting in a smaller centriolar patch surface.

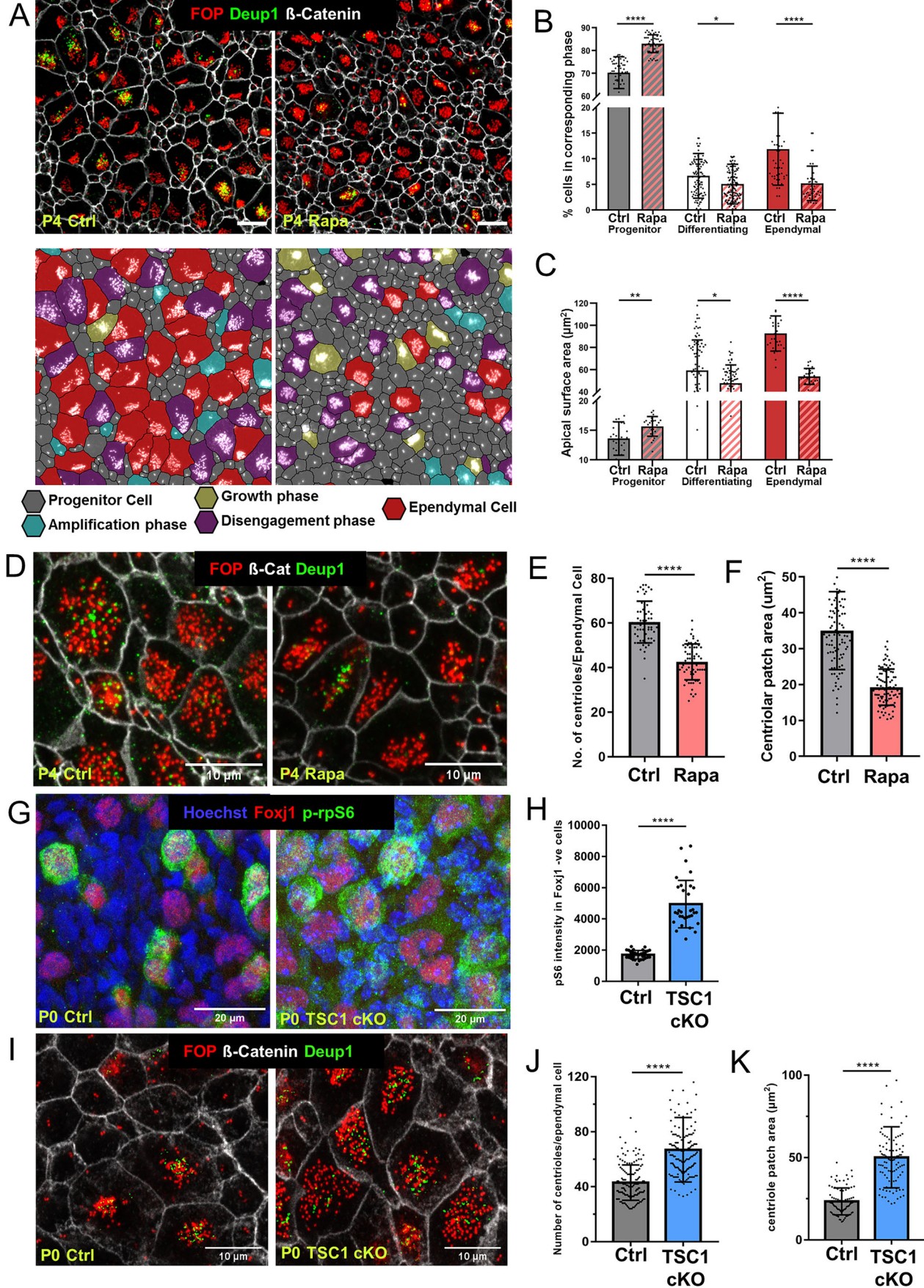

◄ **Figure 2. mTORC1 activity controls ependymal apical cell size and centriole number.**

(A) P4 lateral walls from control and rapamycin-treated pups and their segmentation. Cells are color-coded based on their stage of differentiation. Scale bar: 10 μm. (B, C) Percentage (B) and size of the apical surface (C) of progenitor, differentiating and mature ependymal cells in rapamycin and control conditions; *P = 0.0429 in (B) and 0.0135 in (C); **P = 0.0024 in (C) and ****P < 0.0001 in (B, C). (D) P4 ventricular wall in control and rapamycin-treated pups, showing centrioles with FOP, deuterosomes with Deup1, and cell junctions with ß-catenin. (E, F) Number of centrioles (E) and area occupied by centriolar patch in differentiated cells at P4 (F). (G) En face view showing expression of p-rpS6 and Foxj1 in Control and Tsc1 cKO. (H) pS6 mean gray intensity in Foxj1+ cells in control and Tsc1 cKO. (I) P0 brain lateral ventricular en face labeled with Fop (centrioles, red), b-Catenin (cell junction, white), and deup (deuterosome, green) of control and *Tsc1^{ko/lox}; Nestin-cre* (Tsc1 cKO) mice. (J, K) Number of centrioles and size of the centriolar patch in differentiated cells in Tsc1 cKO and controls at P0. Each dot in the quantifications corresponds to one cell (n > 3); ****P < 0.0001 in (E, F, H, J, K). Data information: In (B, C, E, F, H, J, K), data are presented as mean ± SD. ****P < 0.0001, ***P < 0.001, **P < 0.01, *P ≤ 0.05 (Student's t test). Source data are available online for this figure.

Ependymal cells were also cultured from Tsc1 cKO mice. Regardless of their differentiation stages, p-rpS6 immunostaining was uniform in the cells, and the number of multicentriolar cells increased at Dif 4 but not at Dif 8 compared to controls, suggesting that mTORC1 accelerates differentiation but is not sufficient to induce it in all the cells (Fig. EV4A,B). Similar results were obtained in ependymal cultures prepared from *Tsc1^{lox/lox}* mice infected with a Cre-GFP or GFP expressing adenovirus at Dif 0 and then analyzed at Dif 4 and 8. (Fig. EV4C–E). These results show that mTORC1 activation is necessary to trigger ependymal cell differentiation but not sufficient for their specification.

## mTORC1 controls metabolic rewiring of ependymal cell during differentiation

Given the well-established function of mTORC1 as a master regulator of metabolism, we investigated the metabolic state of ependymal cells undergoing differentiation. We therefore performed a steady-state metabolic profiling using targeted mass spectrometry on ependymal cells at Dif 1 and Dif 4 treated or not with Rapamycin during the last 25 h. We identified 131 metabolites that we analyzed using Principal Component Analysis (Fig. 4A). Of these metabolites, 70 that changed significantly among at least 2 conditions were visualized in heatmaps generated using weighted correlation network analysis (WGCNA) and in volcano plots (Fig. 4B,C; Dataset EV1). Metabolomes at Dif 4 clustered differently from those at Dif 1 and Rapamycin treatment significantly affected this spatial discrimination (Fig. 4A). This indicates that during ependymal differentiation cells rewire their metabolic roadmap and that mTORC1 plays a critical role in this process. Next, we conducted a KEGG-based pathway enrichment analysis among the different metabolomes. This highlighted several pathways, including phospholipid and aspartate metabolisms among the top 5 pathways (Fig. EV5A). Aspartate is an essential precursor of asparagine and pyrimidines synthesis, through asparagine synthetase (ASNS) and carbamoyl phosphate synthetase 2-aspartate transcarbamylase-dihydroorotase (CAD), respectively. Interestingly, our analysis shows that the aspartate/asparagine ratio increases in Dif 4 vs. Dif 1 cells in a Rapamycin-sensitive manner, whereas the aspartate/N-carbamoyl-aspartate ratio sharply goes up after mTORC1 inhibition (Fig. EV5B) mostly due to reduced N-carbamoyl-aspartate levels (Fig. 4D), likely the result of the inhibition of the carbamoyl phosphate synthetase activity of CAD. These results suggest that during ependymal differentiation aspartate is routed towards pyrimidines synthesis through CAD. Moreover, glutamine, the other substrate of the CAD complex, accumulates in Dif 4 cells (Figs. 4D and EV5C). Furthermore, the

levels of Orotic acid, a metabolite further downstream in the pyrimidine synthesis pathway, increases at Dif. 4 vs. Dif 1 in a rapamycin-sensitive manner (Fig. 4D). Overall, the effects of rapamycin on pyrimidine synthesis are reflected on the levels of the nucleosides UMP and UDP (Figs. 4D and EV5D). Interestingly CAD is phosphorylated by S6K1 in the mTORC1 pathway and its activity is positively regulated by mTORC1 to generate N-carbamoyl-aspartate (Ben-Sahra et al, 2013). These data suggest that during ependymal cell differentiation mTORC1 drives the synthesis of pyrimidines precursors of nucleotides essential for glycosylation, RNA, DNA, and phospholipid biosynthesis. The maintenance of elevated rates of pyrimidine synthesis may therefore sustain the biosynthetic demand of ciliary components at several levels, transcriptional and translational to provide ciliary proteins, and metabolic by providing phospholipid precursors such as CTP.

Phospholipid synthesis through the Kennedy pathway requires Cytidine triphosphate (CTP) for the activation of p-Cho and phosphoethanolamine (pEtn) metabolites prior to their condensation with diacylglycerol (DAG) to synthetize phosphatidylcholine and phosphatidylethanolamine, respectively (Fig. EV5E). Interestingly, the p-Cho/CDP-choline ratio, a reliable indicator of the Kennedy pathway flux (Tighanimine et al, 2024), is reduced markedly in cells at Dif 4 vs. Dif 1, an effect reversed by Rapamycin treatment (Fig. EV5F). This suggests that the Kennedy pathway flux is increased during differentiation and might explain the lack of CTP accumulation in cells at Dif 4 due to a higher turnover (Fig. EV5F). Moreover, G3P, the precursor of DAG, accumulates in cells at Dif 4 in a Rapamycin-sensitive manner (Fig. EV5F). Collectively, these data show that ependymal cells differentiation is associated with a profound metabolic reprogramming that relies on mTORC1 for pyrimidines precursors synthesis and downstream phospholipid remodeling likely to support multiciliation and the increase in cell size associated with it.

## S6 kinase inactivation does not abolish rapamycin sensitivity

The experiments above demonstrate that rpS6 phosphorylation and the regulation of pyrimidine biosynthesis correlate with the commitment phases of ependymal cell differentiation. Since S6K1 and S6K2 catalyze rpS6 and CAD phosphorylation in the mTORC1 pathway (Ben-Sahra et al, 2013; Pende et al, 2004), we addressed the functional role by evaluating ependymal differentiation in cells lacking both kinases (S6K−/− cells). As expected, rpS6 phosphorylation was sharply blunted in S6K−/− cells, similar to

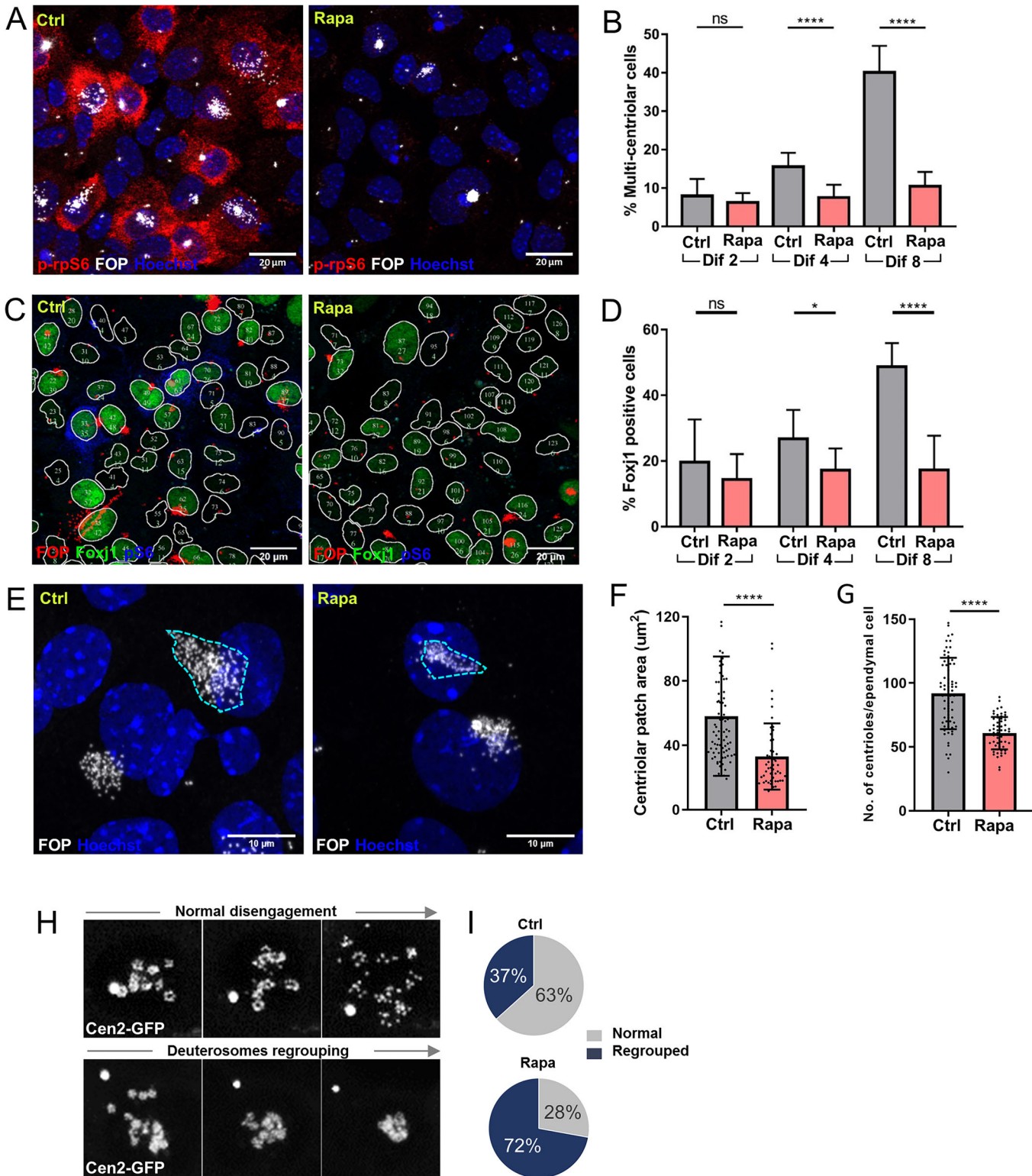

rapamycin-treated cells (Fig. EV4F). However, S6K1/S6K2 deletion had a minor effect on ependymal cell differentiation (Fig. EV4G). S6K−/− cells displayed a similar sensitivity to the inhibitory action of rapamycin on ependymal cell differentiation compared to wild-type cells. Thus, S6K activity on rpS6 phosphorylation is a valuable marker of ependymal cell differentiation, though it is not required for this process, suggesting that additional mTORC1 targets play an important role.

**Figure 3.  mTOR inhibition by rapamycin inhibits ependymal cell differentiation.**

(A) Representative images of primary cells at div 4 immunolabeled with FOP (centrioles) and pS6 antibodies in control and rapamycin conditions. (B) Percentage of multicentriolar cells at different time points in the indicated conditions; ns $P = 0.3684$; ****$P < 0.0001$. (C) Representative segmented images of primary cells at div 4 immunolabeled with FOP and FoxJ1. Each cell mentions its mean gray intensity of the Foxj1 signal. (D) The percentage of Foxj1+ cells is calculated by setting a threshold and relying on automated intensities; ns $P = 0.4150$; *$P = 0.027$; ****$P < 0.0001$. (E) Representative images of primary cells at div 8 immunolabeled with FOP in control and rapamycin conditions. Turquoise boundaries show manual demarcation of the centriolar patch area. (F) The size of the centriole patch area at div 8 in control and rapamycin conditions; ****$P < 0.0001$. (G) Number of centrioles per differentiated cells at div 8 in control and rapamycin condition; one dot corresponds to one replicate; ****$P < 0.0001$. (H) Single-cell monitoring of centriole amplification dynamics in differentiating Cen2GFP+ progenitors between div 4 and 6 in control or the presence of rapamycin. Representative images from three time points showing normal disengagement of centrioles in control (top) and deuterosomes regrouping (bottom) in rapamycin-treated cells. (I) Percentage of cells with regrouped centrioles after the disengagement phase; each dot is one cell. Data information: In (B, D, F, G), data are presented as mean ± SD. ****$P < 0.0001$, *$P \leq 0.05$, ns not significant (Student's $t$ test). Source data are available online for this figure.

## Expression of cell cycle genes and differentiation markers after rapamycin treatment

In many cell types and experimental models, S6K deletion does not mimic the action of rapamycin on cell cycle progression of proliferating cells (Dowling et al, 2010; Montagne et al, 1999; Ohanna et al, 2005). Since many cell cycle proteins have been shown to participate in an alternative cell cycle essential for centriole amplification and multiciliated cell differentiation (Al Jord et al, 2017; Choksi et al, 2024; Khoury Damaa et al, 2025; Ortiz-Alvarez et al, 2022; Serizay et al, 2025), we measured the effect of rapamycin on their expression. As shown in Fig. 5A,B, a 25-h treatment with rapamycin during differentiation was sufficient to reduce the expression of G1 markers (P-Rb, Cyclin D1), G2 markers (Cyclin A, P-Cdk1, Wee1), replication stress markers (p21), and ependymal cell-specific cyclins (Cyclin O). Notably, a subset of these cell cycle proteins was recently shown to be involved in the differentiation of ependymal cells. Cyclin D1 overexpression increased the proportion of multiciliated cells (Choksi et al, 2024), while treatment with CDK4/6 inhibitors (Choksi et al, 2024) or the expression of a non-phosphorylatable Rb (Basso et al, 2024) blocked ciliogenesis. The absence of p21 in progenitor cells hindered ependymal cell differentiation by maintaining cell division (Ortiz-Alvarez et al, 2022). Cyclin O deletion reduced centriole number (Funk et al, 2015; Khoury Damaa et al, 2025). These data suggest that mTORC1 inhibition blunts ependymal cell differentiation, at least in part, by acting on the proteins of the alternative cell cycle for centriole amplification.

## MCIDAS and E2F4 overexpression rescue rapamycin inhibition of multiciliation and upregulate mTORC1

MCIDAS is a master regulator of ependymal cell differentiation, promoting the transcription of genes encoding structural components of centrioles and regulatory proteins (Kyrousi et al, 2015; Ma et al, 2014). MCIDAS lacks motifs associated with DNA binding but is recruited to DNA by forming a complex with the E2F proteins, including E2F4 (Kim et al, 2018). While Rb1 phosphorylation is necessary for E2F release and activation during cell proliferation (Konagaya et al, 2024), it is also important for the activation of ependymal differentiation through MCIDAS (Basso et al, 2024; Choksi et al, 2024). To establish the epistatic interaction between mTORC1 and MCIDAS/E2F4, an adenovirus expressing both MCIDAS and E2F4 proteins was transduced into ependymal cells (Fig. 6A,B). As expected, MCIDAS/E2F4 promoted a two- to three-fold increase in multicentriolar and multiciliated cells

(Fig. 6C,D). While rapamycin reduced the number of differentiated ependymal cells in control cells, MCIDAS/E2F4- overexpressing cells became resistant to the action of rapamycin. These results were also confirmed in the heterologous system of fibroblast cultures, in which MCIDAS/E2F4 expression is sufficient to induce the multiciliation program (Fig. 6E,F) (preprint: Boudjema et al, 2024; Kim et al, 2018). Thus, mTORC1 acts upstream of the MCIDAS/E2F4 program of ciliogenesis. In turn, MCIDAS/E2F4 expression in ependymal cells upregulated mTORC1 activity, as assessed by the phosphorylation of rpS6 and CAD, while down-regulating mTORC2 activity, as assessed by the phosphorylation of Akt and GSK3 (Fig. 6A). These results suggest that mTORC1 acts upstream of MCIDAS/E2F4, probably at the late G1 phase of the alternative cell cycle in differentiating cells. In addition, the positive feed-forward control of MCIDAS/E2F4 on mTORC1 may contribute to the observed activation of this pathway during differentiation (Fig. EV1).

## Phosphoproteomic screening of Dif 4 ependymal cells

To potentially uncover novel mTORC1 targets playing a role in ependymal cell differentiation, we exploited a phosphopeptide enrichment method coupled with liquid chromatography tandem mass spectrometry (LC-MS/MS) (Bonucci et al, 2020). We compared ependymal cell cultures at Dif 4 treated with rapamycin or a control vehicle during the last 24 h. Fe-NTA Immobilized Metal Affinity Chromatography (IMAC) was used to enrich for phosphopeptides. 28474 unique phosphopeptides were identified (Dataset EV2). Notably, many known mTOR substrates were identified and quantified in the analysis, such as S6K1, 4EBP2, LARP1, eIF4G, TIF1A, MITF, AMPK, IRS2, PASK, AS160, Tau, Raptor, PRAS40 (Dataset EV3). Out of 57 mTORC1 substrates reported in a recent review in different cell types (Battaglioni et al, 2022), 14 were significantly downregulated in rapamycin-treated ependymal cells, highlighting the ability to find target phosphorylation sites of interest using this methodology.

Next, we ran a gene ontology (GO) analysis to evaluate whether the differentially expressed phosphopeptides were enriched in specific functional classes relative to the total phosphopeptides detected in each screen. This analysis revealed that the GO terms of cytoskeleton organization, cell cycle and mTOR signaling were significantly overrepresented among the differentially expressed phosphopeptides (Fig. 7A). The consensus motif generated from the differentially regulated phosphopeptides displayed an enrichment of proline and serine, similar to known mTOR substrates (Fig. 7B), suggesting that the phosphoproteins may be direct mTOR

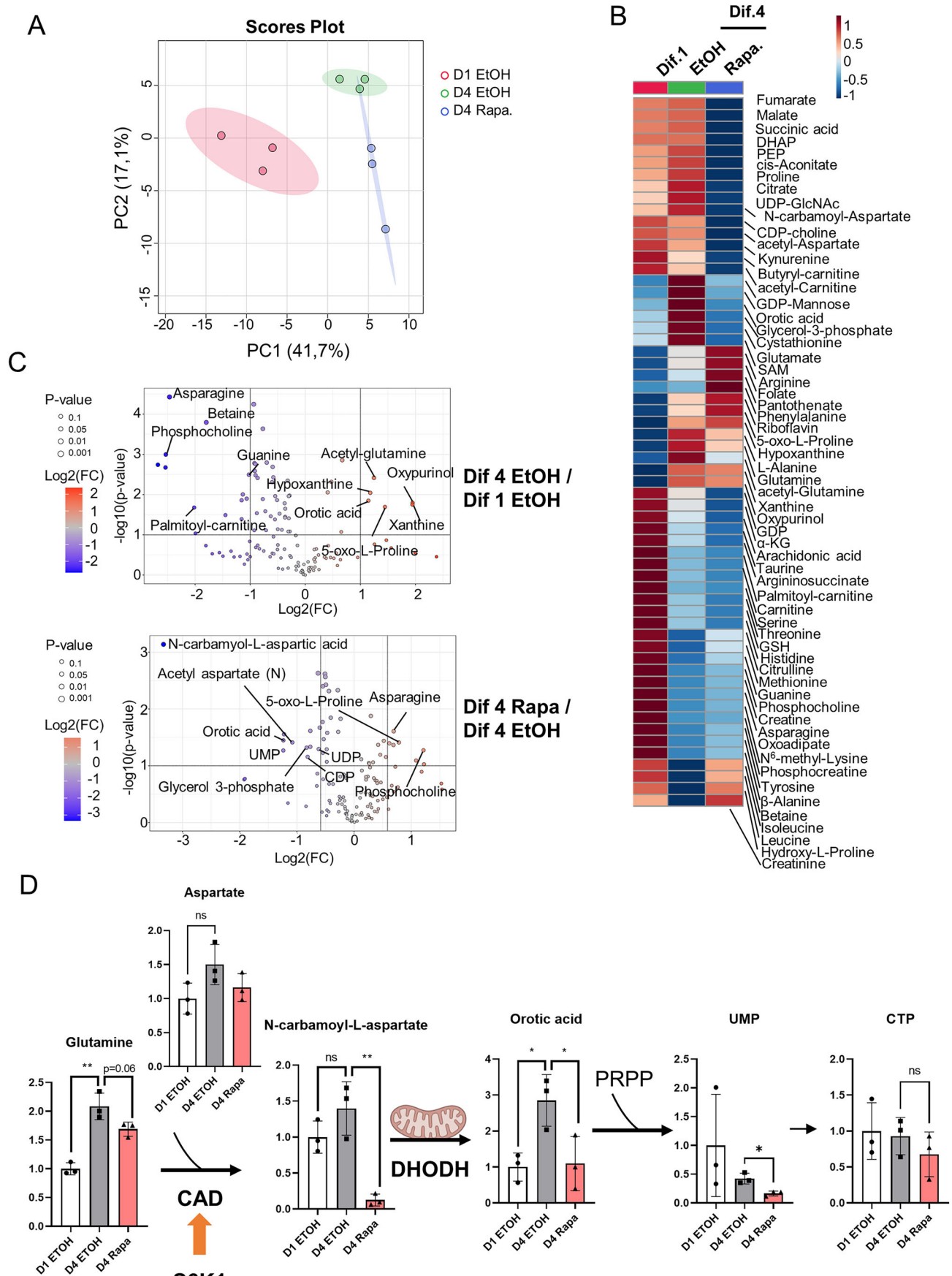

◄ **Figure 4. mTORC1 controls metabolic rewiring during ependymal cell differentiation.**

(A) Metabolome-based principal component analysis (PCA) of the indicated cellular settings. (B) Heatmap showing modules of metabolites in progenitors undergoing differentiation in the presence or the absence of rapamycin using a hierarchical clustering approach. (C) Volcano plots of metabolites differentially modulated in ependymal cells compared to progenitors (top panel) or ependymal cells treated or not with Rapamycin (bottom panel). P value was calculated using Student's t test, n = 3. (D) Summary of the metabolic profiling of the pyrimidine synthesis pathway in ependymal cells at Dif 1 and Dif 4 treated or not with Rapamycin. Data information: The bars represent the mean +/− SEM, each dot is one replicate, n = 3. **P < 0.01, *P ≤ 0.05 (Student's t test).

targets. Among the proteins with a function in the regulation of the cell cycle, cytoskeleton dynamics and ependymal biology were the following: RB transcriptional co-Repressor Like 1 (RBL1, p107), an inhibitor of E2F1-6 transcriptional activity (Beijersbergen et al, 1995); Growth arrest specific 2 like 1 (GAS2L1), a cytolinker protein associated with centrioles (Au et al, 2017; Au et al, 2020; Stroud et al, 2014; van de Willige et al, 2019); Ubiquitin specific peptidase 9 X linked (USP9X), a deubiquitylase regulating ciliogenesis and centriolar satellite integrity (Das et al, 2017; Han et al, 2019; Li et al, 2017); Prune1, a phosphoesterase regulating microtubule polymerization, whose mutations trigger periventricular heterotopia with microcephaly (Zollo et al, 2017); Nuclear Mitotic Apparatus protein 1 (NuMA), a dynein interacting protein involved in spindle assembly (Gallini et al, 2016); Tumor protein translationally controlled (TPT1), a protein involved in the DNA damage response (Jeong et al, 2021); Rho-associated coiled-coil containing protein kinase 1 (Rock1), which promotes actin polymerization (Maekawa et al, 1999) and might be involved in the observed effects of mechanical forces on ependymal differentiation (Basso et al, 2024) (Fig. 7C; Dataset EV4).

The RBL1 phosphopeptide downregulated by rapamycin treatment contains Ser-983 and Thr-992 in a region adjacent to the previously reported Cdk4 phosphorylation sites Ser-959 and Ser-970 (Leng et al, 2002). These data are consistent with the downregulation of RB phosphorylation on the Ser-807 and Ser-811 Cdk sites, as observed by immunoblot analysis (Fig. 5A). Recently, RB family members have been demonstrated to regulate ependymal cell differentiation by acting on E2F transcription factors and MCIDAS (Basso et al, 2024; Quiroz et al, 2024). Together with the resistance of E2F4/MCIDAS overexpressing cells to the inhibitory action of rapamycin on ependymal cell differentiation (Fig. 6), this evidence supports a model by which the phosphorylation of RB family members and the subsequent activation of E2F/MCIDAS is a major regulatory node downstream of mTORC1 activity.

**GAS2L1 phosphorylation promotes centriole dispersion**

Next, we decided to move our attention from cell cycle regulatory proteins to centrosomal-related targets in the mTORC1 phosphoproteome. We asked whether their direct regulation may be involved in the observed centriolar dynamics and spatial regulation (Fig. 3E–I). We focused on GAS2L1 because previous studies in proliferating cells demonstrated its role in centrosome disjunction during cell division (Au et al, 2017; Au et al, 2020). In addition, mutations in genes encoding the homologous proteins Gas2L2 and Gas2L3 in humans or animal models displayed severe phenotypes of primary cilia dyskinesia and brain morphogenesis, suggesting an additional role of this cytolinker protein family in ciliated cells, as well as in brain development (Bustamante-Marin et al, 2019;

Sharaby et al, 2014). The MS analysis identified the GAS2L1 peptide from amino acids 480 to 507 as differentially phosphorylated upon rapamycin treatment (Dataset EV4). This protein region contains a serine and proline-rich sequence consistent with the mTORC1 consensus sequence of known substrates. Surprisingly, rapamycin treatment increased GAS2L1 protein levels in ependymal cells (Fig. 7D,E). However, GAS2L1 protein phosphorylation was decreased by rapamycin treatment, as assessed by radioactive phosphate incorporation, thus confirming the MS analysis (Fig. 7F,G).

To address the role of GAS2L1 in centriolar disjunction, at Dif 0, we transduced cell cultures with wild-type GAS2L1, phosphomutant GAS2L1 in which the serine (S) 482, 489, 493 were mutated to alanine (A), as well as shRNA against GAS2L1 (Fig. 8A). Overexpression of GAS2L1 led to an intense cytoskeletal staining, which was not perturbed by the S to A mutation (Fig. 8B). Both wild-type and phosphomutant GAS2L1 did not perturb the number of multicentriolar cells at Dif 4. At the same time, GAS2L1-shRNA impaired ependymal cell differentiation (Fig. 8C,D). The overall number of centrioles in the multicentriolar cells was not different among the four experimental conditions (Fig. 8E). However, the overexpression of GAS2L1 promoted the disengagement of centrioles in ependymal cells, as shown by the measure of centriole density (Fig. 8F). The phosphomutant and the shRNA could not mimic the effect of wt GAS2L1. These data indicate a role of mTORC1-mediated phosphorylation in centriolar disengagement, contributing to the coordinated action of the mTORC1 pathway on ependymal cell differentiation.

## Discussion

In this study, we reveal the potent action of the mTORC1 pathway on ependymal cell differentiation, from the centriole amplification to their disengagement at the cell surface. We prove that mTORC1 inhibition maintains the pool of ependymal progenitors by keeping a quiescent state and blocking the alternative cell cycle progression essential for centriole amplification. Forcing E2F4 and MCIDAS expression bypasses this block and triggers ciliogenesis even in the presence of rapamycin, concomitantly sustaining mTORC1 activation in a positive feed-forward mechanism. mTORC1 inhibition in multiciliated cells affects centriole disengagement, suggesting a direct action of mTORC1 on centrosomal dynamics. By phosphoproteomic analysis and phosphomutant expression, we identify GAS2L1 phosphorylation as a novel mTORC1 target involved in this process.

While serving as a master regulator of cell mass accumulation and size in all eukaryotic cells, both proliferating and terminally differentiated, the mTOR pathway also cross-talks with cell cycle proteins to coordinate growth and cell division in cycling cells

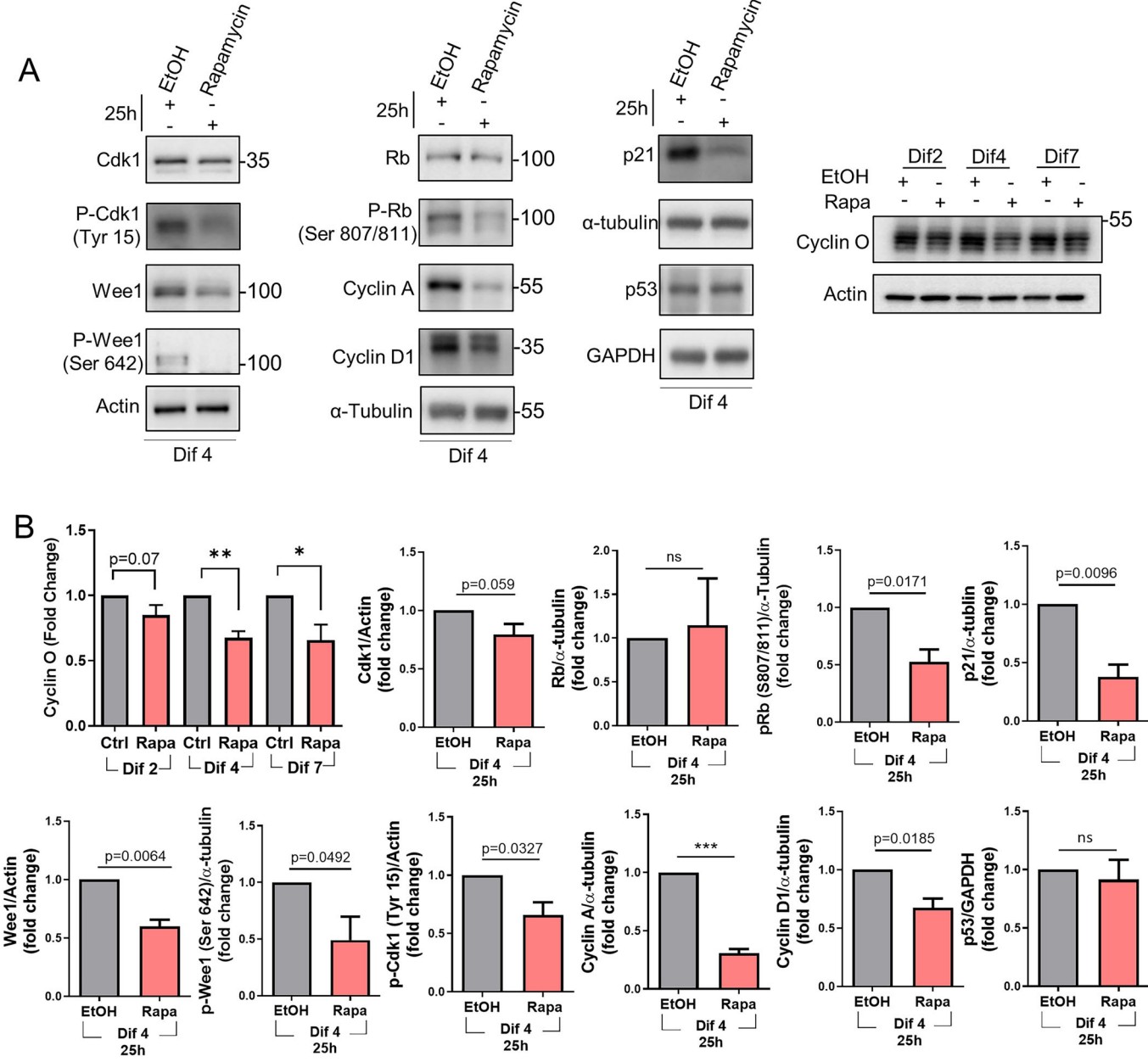

**Figure 5. Expression of cell cycle genes after rapamycin treatment.**

(A) Cells at dif 3 were treated 25 h with vehicle (EtOH) or with Rapamycin at 20 nM, then proteins were extracted for western blot analysis, and extracts were probed with the indicated antibodies. (B) Fold change of EtOH-treated cells over Rapamycin-treated cells of the indicated proteins relative to the indicated loading control. Data information: The bars represent the mean $+/-$ SEM, $n = 3$. $P$ values were determined using a two-sided Student's $t$ test. $*P = 0.0380$, $**P = 0.0077$, $***P = 0.0009$. Source data are available online for this figure.

(Liu and Sabatini, 2020; Rubin et al, 2020). mTORC1 activity oscillates throughout the different phases of the cell cycle and acts on multiple cyclins and Cdk-inhibitors controlling Cdks (Joshi et al, 2024). mTORC1 has been reported to regulate cell cycle proteins at various levels through transcription, translation, and nuclear import. At the G1/S phase transition, Myc and FOXK1 are transcription factors whose activity on cell cycle progression is influenced by mTORC1 (He et al, 2018). In addition, mTORC1-mediated 4EBP phosphorylation participates in the translational

regulation of cyclin D1 and other cell cycle protein-encoding mRNAs (Dowling et al, 2010). The additional mTORC1 target, SGK1, regulates p27 phosphorylation and nuclear import (Hong et al, 2008). mTOR also influences Wee levels, with contrasting results on the G2/M transition depending on the experimental models (Atkin et al, 2014; Joshi et al, 2024). We show that the alternative cell cycle during ependymal cell differentiation, leading to centriole amplification rather than cell division, is also strongly impacted by mTORC1 inhibition, with the downregulation of both

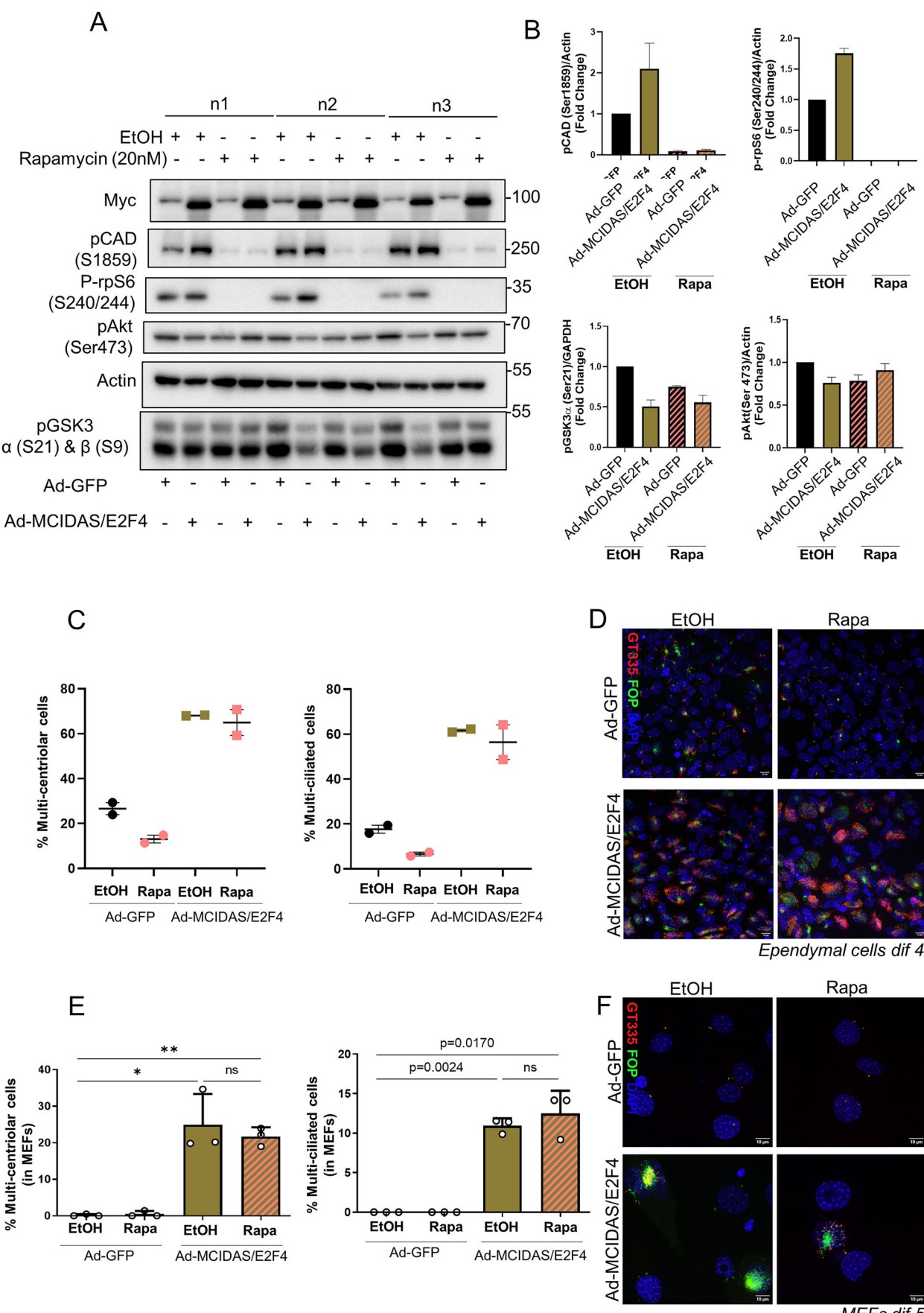

**Figure 6. MCIDAS and E2F4 overexpression upregulates mTORC1 and drive rapamycin-insensitive multiciliogenesis in both ependymal cells and fibroblasts.**

(A) Immunoblot analysis of ependymal cells at Dif 4 after adenoviral transduction with MCIDAS/E2F4 or GFP control virus. (B) Fold change of MCIDAS/E2F4 or rapamycin-treated cells over control ependymal cells of the indicated proteins relative to the indicated loading control. The bars represent the mean $+/-$ SEM, $n = 3$. (C) Percentage of multicentriolar and multiciliated ependymal cells in the indicated conditions. The bars represent the mean $+/-$ SEM, $n = 2$. (D) Representative images of ependymal cells at dif 4 immunolabeled with FOP (centrioles) and GT335 (cilia) in adenoviral GFP or MCIDAS/E2F4 or rapamycin conditions. (E) Percentage of multicentriolar and multiciliated mouse embryonic fibroblasts in the indicated conditions. The bars represent the mean $+/-$ SEM, $n = 3$. P values were determined using a two-sided Student's t test. *$P < 0.05$ and **$P < 0.01$. (F) Representative images of mouse embryonic fibroblasts immunolabeled with FOP (centrioles) and GT335 (cilia) in adenoviral GFP or MCIDAS/E2F4 or rapamycin conditions. Source data are available online for this figure.

G1 and G2 phase markers. S6K activity, which is mainly involved in cell size control downstream of mTORC1, plays a minor role during ependymal differentiation in the face of this pleiotropic action of mTORC1 on cell cycle proteins. Since recent studies have demonstrated that several cell cycle proteins impact centriole amplification (Choksi et al, 2024; Khoury Damaa et al, 2025; Ortiz-Alvarez et al, 2022; Serizay et al, 2025; Vladar et al, 2018), their regulation by mTORC1 likely explains the potent role of rapamycin during early ependymal development. Significantly, overexpression of E2F4 and MCIDAS rescues the rapamycin-induced block of centriolar amplification, consistent with mTORC1 acting upstream of the E2F family of transcription factors. By both immunoblot and phosphoproteomics analysis, we consistently provide evidence that the phosphorylation of RB and RBL1 on CDK-dependent sites important for their inactivation is downregulated by rapamycin treatment. Since the RB and RBL1 targets are the E2F1-6 transcription factors which are required for the MCIDAS-dependent ciliogenesis program (Basso et al, 2024), our data define an epistatic interaction between mTORC1, RB/RBL1, E2F factors, and MCIDAS in the control of ependymal cell differentiation (Fig. 8G). It is likely that the downregulation of G2 proteins observed after rapamycin treatment is a consequence of this effect on G1 phase markers.

The ciliogenesis triggered by MCIDAS and E2F4 is accompanied by further mTORC1 activation. This up-regulation may concur in the striking mTORC1 activation that we observe during the commitment phase of ependymal differentiation, suggesting the existence of a feed-forward mechanism. We can only speculate on the molecular mechanisms underlying the increased mTORC1 activation at this stage. Growth factor receptors and nutrient sensors are essential for mTORC1 activation and are known to concentrate in cilia (Liu et al, 2021). We cannot exclude that centriole amplification is also accompanied by a remodeling of lysosomes, where mTORC1 activation occurs, as recently proposed (Kimura et al, 2021). To explain the shut-off of mTORC1 activity in terminally differentiated cells, a possible action of cilia bending on mTORC1 activity could be evoked, as previously observed for primary cilia of kidney cells sensing urinary flow (Boehlke et al, 2010). Future studies should address these possibilities.

The profound effect of mTORC1 on ependymal cell differentiation is underscored by its ability to regulate centrosomal proteins, such as GAS2L1 (Fig. 8G). Rapamycin inhibits GAS2L1 phosphorylation in the 480-507 aa region, which contains several mTOR-like proline-directed consensus sites for serine phosphorylation. GAS2L1, as its homologs GAS2L2 and GAS2L3, are centrosomal proteins able to interact with microtubule plus-end tracking proteins EB1, EB2, and EB3 (Stroud et al, 2014). In addition, they have a cytolinker activity and, therefore, provide a molecular bridge

with both actin and microtubule cytoskeletons that exert the forces required to split the centrosomes into proliferating cells (van de Willige et al, 2019). Silencing GAS2L1 expression inhibits centrosome movement and disjunction, while GAS2L1 overexpression drives premature separation of the centrioles (Au et al, 2017). Here, we show that a similar role of GAS2L1 underlies the centriolar disengagement during ependymal cell differentiation and is influenced by mTORC1-mediated phosphorylation. The mTOR phosphorylation sites are located between the N-terminus region containing the calponin homology (CH) and the Gas2-related (GAR) domains for microtubule and actin binding, respectively, and the C-terminus region containing the SxIP motif for EB binding (Stroud et al, 2014). The fact that Gas2L family members have essential roles in proliferating and multiciliated cells is further supported by human genetics, revealing the loss of function of Gas2L2 mutations in patients suffering from primary cilia dyskinesia (Bustamante-Marin et al, 2019). In this disease, motile cilia dysfunction results in defective mucociliary clearance and chronic airway pathologies, such as respiratory distress, bronchiectasis, and chronic rhinosinusitis. It will be essential to evaluate potential defects in GAS2L1 function in conditions altering the ependyma, which contains motile cilia.

Rapamycin treatment has been approved for treating subependymal giant cell astrocytomas (SEGAs), benign tumors that may develop in TSC patients and cause hydrocephaly by ventricle obstruction (Franz et al, 2014). In addition, rapamycin promotes longevity and counteracts senescence and age-related functional impairment in a broad range of aging models (Harrison et al, 2009; Liu and Sabatini, 2020). Describing the metabolic and phosphoproteomic landscape sensitive to rapamycin in neural progenitors and during ependymal differentiation will help define additional molecular mechanisms well beyond the proof-of-concept analysis of GAS2L1.

## Methods

### Reagents and tools table

| Reagent/resource | Reference or source | Identifier or catalog number |
|---|---|---|
| **Experimental models** | | |
| OF1 (*M. Musculus*) | Charles River Lab | |
| Centrin2-GFP (*M. Musculus*) | Jackson Lab | CB6-Tg(CAG-EGFP/CETN2)3-4Jgg/J |
| Nestin-Cre (*M. Musculus*) | Tronche et al, 1999 | |

| Reagent/resource | Reference or source | Identifier or catalog number |
| --- | --- | --- |
| S6K1/S6K2 (*M. Musculus*) | G. Thomas lab | Pende et al, 2004 |
| Tsc1 lox (*M. Musculus*) | JAX | #005680 |
| Tsc1 ko (*M. Musculus*) | JAX | #005680 |
| **Recombinant DNA** | | |
| **Antibodies** | | |
| Mouse anti-FoxJ1 | ThermoFisher Scientific | 14-9965-82 |
| Mouse anti-GT335 | Adipogen | AG20B0020C100 |
| Mouse anti-p27 | BD Biosciences | 610241 |
| Mouse anti-β Catenin | Millipore | MAB2081 |
| Mouse anti-FOP | Abnova | H00011116- M01 |
| Mouse anti-Sas6 | Santa Cruz Biotech | Sc-81431 |
| Mouse anti-DM1a | SIGMA | T6199 |
| Mouse cdc2 p34 | Santa Cruz Biotech | Sc-54 |
| Mouse anti-cyclin D1 | Santa Cruz Biotech | Sc-450 |
| Mouse anti-p21 | Santa Cruz Biotech | Sc-6246 |
| Mouse anti-β-Actin | Proteintech | 66009-1-Ig |
| Mouse anti-α-Tubulin | SIGMA | T9026 |
| Mouse anti-Myc-Tag | Cell Signaling Technology | 2276 |
| Mouse anti-S6 | Cell Signaling Technology | 2317 |
| Rabbit anti-p-GSKα/β | Cell Signaling Technology | 9331 |
| Rabbit anti-p-Akt | Cell Signaling Technology | 4060 |
| Rabbit anti-Akt | Cell Signaling Technology | 9272 |
| Rabbit anti-pS6K1 | Cell Signaling Technology | 9205 |
| Rabbit anti-S6K1 | Cell Signaling Technology | 2708 |
| Rabbit anti-p-CAD | Cell Signaling Technology | 12662 |
| Rabbit anti-CAD | Cell Signaling Technology | 93925 |
| Rabbit anti-cyclin O | Invitrogen | PA5-20263 |
| Rabbit anti-GAPDH | Cell Signaling Technology | 2118 |
| Rabbit anti-p53 | Santa Cruz Biotech | Sc-6243 |
| Rabbit anti-cyclin A | Santa Cruz Biotech | Sc-596 |
| Rabbit anti-p-Rb | Cell Signaling Technology | 9308 |
| Rabbit anti-Rb | Santa Cruz Biotech | Sc-50 |

| Reagent/resource | Reference or source | Identifier or catalog number |
| --- | --- | --- |
| Rabbit anti-p-wee1 | Cell Signaling Technology | 4910 |
| Rabbit anti-wee1 | Cell Signaling Technology | 13084 |
| Rabbit anti-p-cdc2 | Cell Signaling Technology | 9111 |
| Phalloidin-Actin-488 | Invitrogen | A12379 |
| Rabbit anti-p73 | Abcam | ab40658 |
| Rabbit anti-p-rpS6 | Invitrogen | 44-923G |
| Rabbit anti-p-rpS6 | Cell Signaling Technology | 5364 |
| Rabbit anti-p27 | Santa Cruz Biotech | Sc-528 |
| Rabbit anti-Deup1 | Homemade | Mercey et al, 2019 |
| Rabbit anti-Gas2l1 | Sigma | HPA019858 |
| Alexa Fluor-conjugated secondary antibodies | Invitrogen | |
| **Oligonucleotides and other sequence-based reagents** | | |
| **Chemicals, enzymes, and other reagents** | | |
| Papain | Worthington | 3126 |
| DNase1 | Worthington | 2139 |
| Trypsin inhibitor | SIGMA | 10109878001 |
| Trypsin | Gibco | 25300-054 |
| Poly-L-lysine | SIGMA | P1524 |
| Rapamycin | Merck | RO395 |
| DMSO | Merck | 1029311001 |
| Triton X-100 | SIGMA | T8787 |
| **Software** | | |
| ImageJ | Open source software | |
| GraphPad Prism 7 | Graphpad software | |
| ZEN software | Zeiss | |
| Metamorph Software | Molecular devices | |
| **Other** | | |
| inverted Zeiss® Axio Observer.ZI™ microscope | Zeiss | |
| inverted microscope Zeiss® LSM 880 Airyscan™ | Zeiss | |
| inverted spinning-disc Nikon® Ti PFS microscope | Nikon | |
| Leica Confocal SP8-gSTED | Leica | |

## Mice

Mice were bred, handled, and utilized for experiments following French and European Union regulations and guidelines of the local ethics committee (comité d'éthique en experimentation animal n°005).

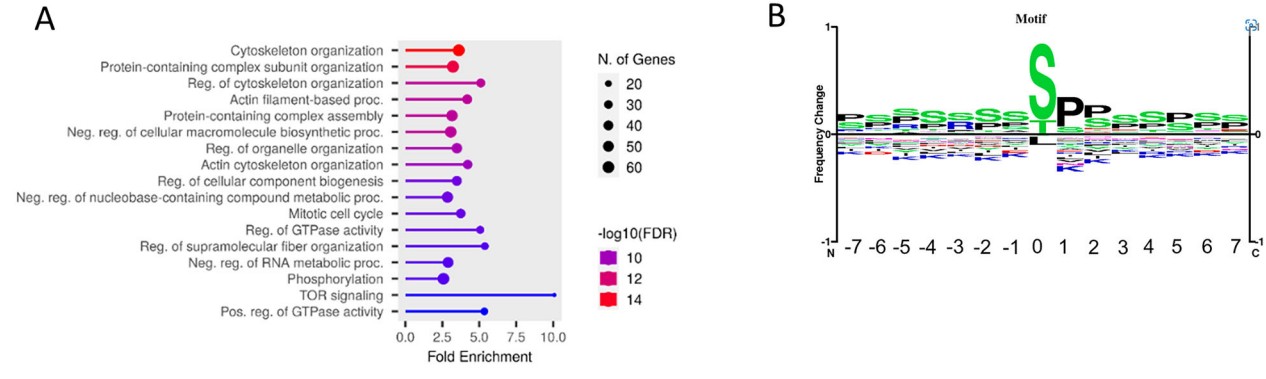

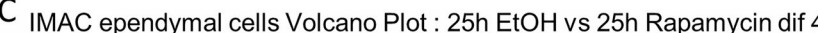

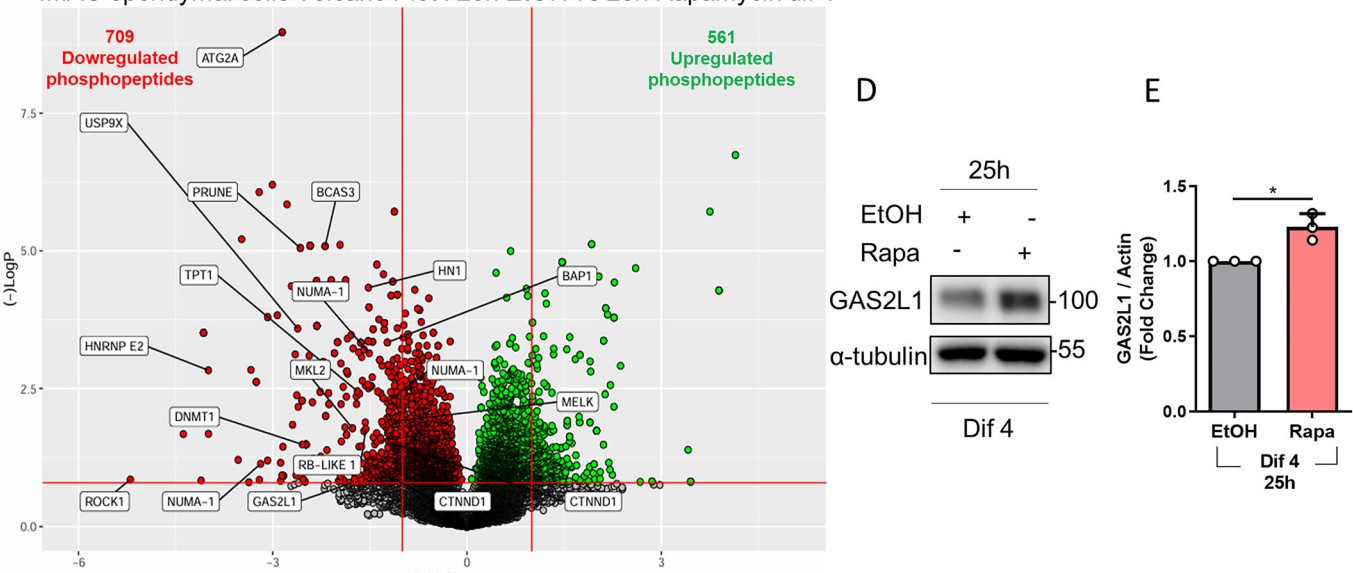

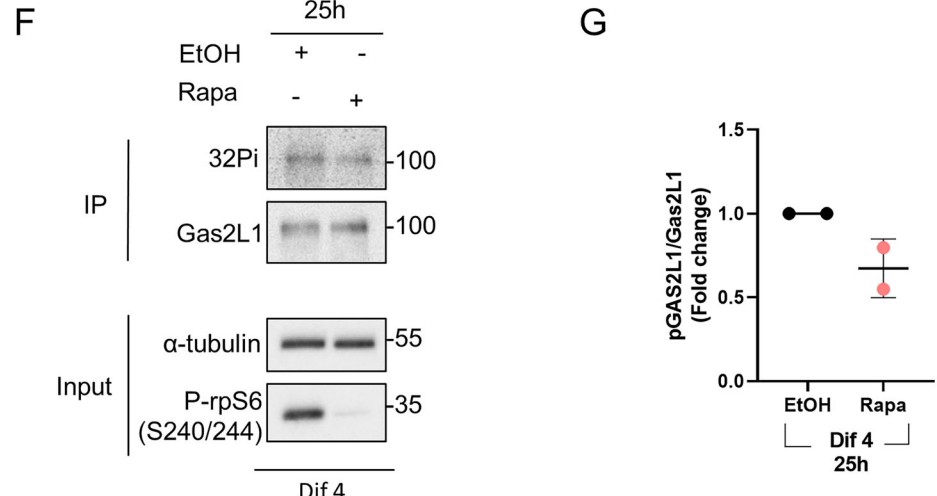

OF1 (Oncins France 1; Charles River Laboratories) strain of mice were used as wild-type models. Other genetic models used were Centrin2-GFP (CB6-Tg(CAG-EGFP/CETN2)3-4Jgg/J; The Jackson Laboratories) (Higginbotham et al, 2004), Tsc1<sup>fl/fl</sup> (The Jackson Laboratories®), Nestin-Cre (Tronche et al, 1999) and S6K1/S6K2-double mutant mice (Pende et al, 2004). Tsc1 cKO was generated by crossing Tsc1<sup>ko/+</sup> and Nestin-Cre mice to get double heterozygous and back-crossing the double heterozygous mice with Tsc1<sup>fl/fl</sup> mice. No

◄ **Figure 7. Phosphoproteomic screening of div 4 ependymal cells treated 25 h EtOH versus div 4 ependymal cells treated 25 h Rapamycin.**

(A) Gene ontology analysis, using ShinyGo, shows statistically significant gene set enrichments by comparing the IMAC statistically significant differentially phosphorylated proteins versus the entire IMAC enrichment set. (B) Consensus motif generated from all the statistically significant differentially phosphorylated proteins. (C) Volcano plots of IMAC enrichment. The x axis shows the log2-ratio for phosphopeptides between samples and the y axis shows the −log10 P value. The horizontal red bar represents the statistical cut-off of significance (P value of 0.05) using Student's t test, n = 2. Red dots represent downregulated phosphopeptides in the 25 h Rapamycin-treated cells and green dots represent upregulated phosphopeptides. Representative peptides are shown. (D) Cells at dif 3 were treated 25 h with vehicle (EtOH) or with Rapamycin at 20 nM then proteins were extracted for Western Blot analysis, and these extracts were probed with GAS2L1 antibodies. (E) Densitometry quantification of the immunoblot analysis. The bars represent the mean $+/-$ SEM, n = 3. P values were determined using a two-sided Student's t test. *P < 0.05. (F) In vivo labeling, less radioactive phosphates are incorporated when cells are treated 25 h with rapamycin. (G) Scatter plot of the two experiments of in vivo labeling. Source data are available online for this figure.

difference between male and female mice was observed in our studies. Therefore, both sexes were treated as equivalent and used indiscriminately for our experiments.

## Ventricular whole mount preparation

To visualize ependymal cells on the lateral walls of the brain's lateral ventricles (in vivo), whole-mount brain ventricles were prepared as previously described (Delgehyr et al, 2015). Mice were sacrificed, and their brains were extracted. Mice older than postnatal day 4 (P4) were first perfused with 4% paraformaldehyde (PFA) before brain extraction. The brains were then dissected under a dissection microscope in 1× PBS. First, the telencephalon was separated from the rest of the brain. The two hemispheres were then halved, and the olfactory bulb, thalamus, and choroid plexus were removed from each hemisphere. The remaining tissue was fixed in either 4% PFA (Electron Microscopy Sciences® 15710; in 1× PBS) at 4 °C for 45 min with gentle shaking or in absolute methanol at −20 °C for 45 min. Following fixation, the tissue was immunostained. The lateral walls of the lateral ventricles were intricately dissected under a dissection microscope, resulting in a thin, intact layer. This flat piece of tissue was then mounted on a glass slide using Fluoromount-G™ mounting solution (ThermoFisher® 00-4958-02).

## MEF cell culture

Mouse embryonic fibroblasts (MEFs) has been purchased from ATCC® (SCRC-1040). MEFs were grown in DMEM (Gibco® 41965039) supplemented with 10% fetal bovine serum (FBS) and 1% penicillin–streptomycin (PS) (Gibco® 15140122) in a 37 °C humidified incubator with 5% $CO_2$. Only MEFs that are maintained in culture for less than six passages were used in the experiments reported here.

## Ependymal cell primary culture

Ependymal cell culture was performed as previously described by our lab (Delgehyr et al, 2015). Neural stem cells were extracted from the brain ependyma of neonatal mouse pups (preferably P0 to P2) and cultured to generate ependymal cells. The pups were sacrificed, and their brains were extracted and kept on ice-cold Hank's solution (1× Hank's Balanced Salt Solution, Gibco® 14060-040; 100 mM Hepes, Gibco® 15630-056; 0.75% Sodium bicarbonate, Gibco® 25280; 100 U/ml penicillin–streptomycin, Gibco® 15140) to maintain cellular viability. Under a dissecting microscope, the ependyma was extracted by removing as much unwanted tissue as possible, including the thalamus, olfactory bulbs, choroid

plexus, meninges, and most of the cortex. The remaining tissue was chopped into small pieces, small enough to be aspirated through 1000-μl pipette tips. To digest the tissue into a single-cell suspension, it was incubated in digestion medium for 1 h at 37 °C (1 mL per brain; 1× DMEM, Gibco® 31966; 1.2 mg/mL Papain, Worthington® 3126; 150 μg/mL DNase1, Worthington® 2139; 288 μg/mL Cysteine, Sigma® C7352). The tissue was centrifuged for 1 min at $100 \times g$, and the supernatant was removed. The digestion reaction was stopped by adding stop solution (1 mL per brain; 1× Leibovitz's L-15 medium, Gibco® 11415; 1 mg/mL Trypsin inhibitor, Sigma-Aldrich® 10109878001; 150 μg/mL DNase1, Worthington® 2139) to the tissue and gently disturbing the pellet by flicking the tube. The tissue was centrifuged for 1 min at $100 \times g$, and the supernatant was removed. The tissue was washed twice with Leibovitz's L-15 medium (ThermoFisher® 11415064). Now, in tiny chunks, the tissue was broken down into a single-cell suspension by gentle pipetting (no more than ten times). The cell suspension was then seeded in Poly-L-Lysine-coated cell culture flasks with 5 mL of growth medium (1× DMEM, Gibco® 31966; 10% decomplemented fetal bovine serum, Gibco® 10270; 100 U/mL penicillin–streptomycin, Gibco® 15140). Each flask contained cells from one brain and was incubated for 5 days at 37 °C, with 5% $CO_2$ and 80–90% humidity (Heraeus® HeraCell™ series: 40161669). The medium was rinsed once after 24 h to remove cellular debris. After 5 days, the culture reached ~90% confluence. The flasks were shaken overnight at 250 rpm (New Brunswick Scientific® Excells E2 platform shaker) at room temperature to remove differentiated cells. The flasks were then rinsed three times with 1× PBS (Eurobioscientific® GAUPBS00-07) to remove cellular debris. The cells were trypsinized (1 mL/T-25 flask; Trypsin, Gibco® 25300-054) for 5 min at 37 °C and collected in a Falcon tube. An equal volume of decomplemented FBS was added, and the cells were centrifuged at $100 \times g$ for 5 min. The supernatant was removed, and the cellular pellet was gently dissociated in the growth medium. The medium volume was adjusted to achieve a concentration of 7.5 million cells per ml. In total, 20 μL of this dense cellular suspension was seeded in the center of each Poly-L-Lysine-coated coverslip placed in a 24-well plate. The plate was gently moved to a cell culture incubator for 60 min to allow cell adhesion. Then, 500 μL of growth medium was gently added to each well from the walls to minimize disturbance. The plate was incubated for another 24 h, after which the wells were rinsed twice with differentiation medium (DMEM, 0% FBS, 1% penicillin–streptomycin) and topped with 500 μL of differentiation medium per well. This marked the initiation of neural stem cell differentiation into ependymal cells, considered 'differentiation day 0' (dif 0).

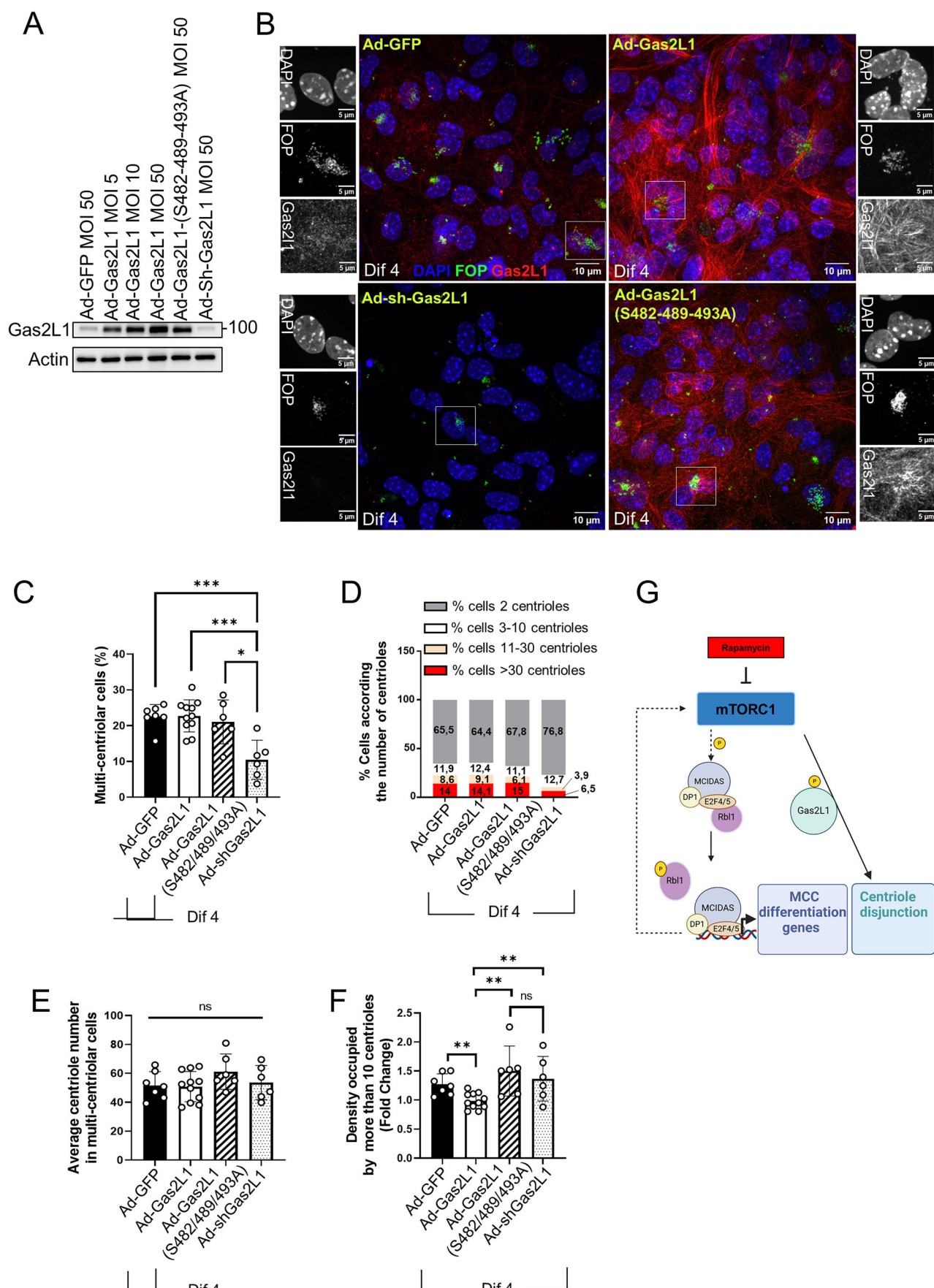

◄ **Figure 8. GAS2L1 phosphorylation promotes centriolar disengagement.**

(A) Immunoblot analysis of ependymal cells at dif 4 after adenoviral transduction with the indicated vectors. (B) Representative images of ependymal cells at dif 4 immunolabeled with FOP (centrioles) and GAS2L1 after adenoviral transduction as indicated. (C, D) Percentage of ependymal cells according to the centriole number in the indicated conditions. $****P \leq 0.0001/***P \leq 0.001/**P \leq 0.01/*P \leq 0.05$ (Student's $t$ test). (E) Average centriole number in multiciliated cells of the indicated conditions. (F) Number of centrioles per ependymal surface. The bars represent the mean $+/-$ SEM, $n > 5$ of independent experiments in which at least 200 cells were counted. $P$ values were determined using a two-sided Student's $t$ test. $*P < 0.05$ and $**P < 0.01$. (G) Schematic representation of the proposed model for mTORC1 regulation of ependymal cell differentiation. Source data are available online for this figure.

## In vitro adenoviral transduction

Adenovirus containing mouse GAS2L1, mouse GAS2L1(S482/489/493 A), GFP-U6-mouse GAS2L1-shRNA, GFP, Cre-GFP were produced by Vector Biolabs®, and the adenovirus containing NLS-6xMyc-mE2F4vp16-S2-MCIDAS is described in (Kim et al, 2018) and was produced by the VVTG platform (SFR Necker, France). The adenoviral transduction in MEFs was previously described (Kim et al, 2018). The adenoviral transduction in ependymal cells is done the day after the overnight flask shaking. Briefly, the cells are washed once with basal medium (DMEM, 0% FBS, 0% PS), then incubated with the diluted virus in basal medium according to the desired MOI during 2 h. For immunoblot analysis, basal medium with virus is discarded and replaced by differentiation medium, the cells are kept in flasks until the differentiation date chosen and then proteins are extracted. For immunofluorescence analysis, after the virus incubation, the basal medium with virus is replaced by growth medium, and the flasks are incubated 24 h. Then, cells are trypsinized as described in the previous paragraph and placed on 24-well plate for immunostaining.

## Poly-L-lysine coating

Flasks were coated with Poly-L-lysine by adding a 2 mL solution (40ug/ml Poly-L-lysine hydrobromide, Sigma P1524), gently shaking to cover the surface with solution, and incubating at 37 °C for 60 min. The PLL solution was removed, and the flasks were washed three times with autoclaved distilled water. Uncapped flasks were dried under a cell culture hood for 1 h before use. A similar procedure was performed for coating glass coverslips (∅ 10 mm) placed in wells of 24-well plates.

## Rapamycin administration in mice

Rapamycin administration in mouse pups was performed daily from P0 to P3. They were placed on ice flakes for 60 s to immobilize and numb the pups. Rapamycin (Merck®, Catalog: RO395) was injected subcutaneously using a 0.5 mL insulin syringe at a 2 mg/kg body weight concentration, with a maximum volume of 50 μL per injection. Immediately after injection, the pups were placed on a heat pack maintained at ~37 °C. A 0.9% NaCl saline solution was used as the solvent for the injection solution. DMSO (Merck®, 1029311001) was used for negative control subjects to replace the rapamycin stock solution.

## In vitro drug treatment on ependymal cell

Rapamycin was added to the ependymal cell culture at dif 0 (differentiation day 0) at a final concentration of 20 nM. An equivalent volume of DMSO or ethanol was added to the negative control group. Acute rapamycin treatment was given at a final concentration of 1 μM on dif-4 cells 2 h before acquiring the first set of images for time-lapse acquisitions of differentiating ependymal cells. Where indicated, ependymal cell cultures at dif 3 were treated for 25 h with Rapamycin at a final concentration of 20 nM.

## Immunostaining

Methanol-fixed samples were used for stainings involving anti-Deup1 and anti-GAS2L1 labeling; for all other combinations, PFA fixation was used. Fixed samples were incubated in saturation medium (10% FBS; 0.2% Triton X-100 Sigma® T8787 in 1× PBS) for 45 min with gentle shaking. All antibody dilutions were prepared in a saturation medium. For primary antibody labeling, samples were incubated overnight at 4 °C. Samples were given three washes with saturation medium at room temperature with gentle shaking. For secondary antibody labeling, samples were incubated for 1 h at room temperature. Samples were washed three times using a saturation medium before mounting. Primary antibodies used were: Anti-Foxj1 mouse IgG1 monoclonal (1:400; ThermoFisher Scientific®, catalog: 14-9965-82, clone: M1/69, lot: 4345885), Anti-GT335 mouse IgG1 monoclonal (1:1000; Adipogen®, catalog AG20B0020C100, lot: A20631002), Anti-p27 mouse IgG1 monoclonal (1:200; BD Biosciences®, catalog: 610241, lot: 4345564), Anti-Sas6 mouse IgG2b monoclonal (1:700; Santa Cruz Biotech®, catalog: sc-81431), Anti-α Tubulin (DM1a; 1:500, SIGMA, T6199), Anti-p73 rabbit monoclonal (1:200; Abcam®, catalog: ab40658, lot: GR249608-6), Anti-p-rpS6 rabbit IgG polyclonal (1:500; Invitrogen®, prod: 44-923 G, lot: 1913151), Anti-p-rpS6 (Ser240/244) rabbit IgG monoclonal (1:800; Cell signaling Technology®, catalog: 5364), Anti-p27 rabbit IgG polyclonal (1:200; Santa Cruz Biotech®, catalog: sc-528, lot: K0912), Anti-ß-Catenin mouse IgG1 monoclonal (1:500; Millipore®, catalog: MAB2081, lot: 2982442), Anti-FOP mouse IgG2b monoclonal (1:1000; Abnova®, catalog: H00011116- M01, lot: L5311-2B1), Anti-Deup1 (1:5000; Homemade, Mercey et al 2019), Anti-GAS2L1 rabbit IgG polyclonal (1:600; Sigma®, catalog: HPA019858, lot: 000056867), Phalloidin-Actin-488 (Invitrogen, A12379). For secondary antibody labeling, species-specific Alexa Fluor fluorophore-conjugated secondary antibodies (1:400, Invitrogen®) were used, accompanied by Hoechst 33342 (1:2000 from 20 mg/mL stock; Sigma Aldrich®, catalog: 14533).

## Imaging

High-magnification images were taken using an inverted Zeiss® Axio Observer.ZI™ epifluorescence microscope, using an apochromat 40× and 63×1.4 NA objective and a Zeiss® Apotome™ with an H/D grid. Confocal images were acquired using an inverted microscope Zeiss® LSM 880 Airyscan™, with 440, 515, and 560

lasers attached. Zen®- provided software was used for both of these microscopies. Time-lapse pictures were taken with an inverted spinning-disc Nikon® Ti PFS microscope, oil-immersion 63× (NA 1.32), evolve EMCCD camera, dpss lasers (491 nm, 10–20% intensity, 100–1000 ms exposition), Metamorph Nx software. Images of sections in low magnification were taken with Leica® MZ16F and attached to Nikon® DS-Ri1 colored camera, LOT® HXP120 lamp, and Nikon-supplied software.

## Ependymal cell segmentation

High-definition images of ventricular whole-mounts comprising labeling of cell junction (β-catenin), centrioles (FOP or centrin), and deuterosomes (deup1 or sas6) were acquired. 2D images of the ventricular surface were extracted from the 3D stack by dedicated computational means developed by our lab (Shihavuddin et al, 2017). A bioinformatics tool devised by our lab would segment the cells based on their cellular junction labeling. A subset of 2000 cells was manually segregated into the five stages of ependymal differentiation, and this was used to train a deep neural network to automate the recognition and segregation of the cells for our data. The tool would also calculate the area of the apical surface of each cell. GAS2L1 images has been observed using Zeiss Spinning Disk and Leica Confocal SP8-gSTED, both using a 63X oil-immersion objective and each software provided by Zen® and Leica®, respectively.

## Classification of differentiating ependymal cells

Classification of ependymal cells into progenitors, fully differentiated ependymal cells, and intermediate stages (growth phase, amplification phase, or disengagement phase) were based upon their centriole–deuterosome morphology as previously described (Spassky and Meunier, 2017).

## Measurement of signal intensity

Z-projection images were acquired using identical parameters to measure a labeling signal intensity, which was acquired using identical parameters. For p-rpS6 intensity measurement, Foxj1-negative cells were manually identified based on the absence of a nuclear Foxj1 signal. Using the "freehand selection" tool in ImageJ, an approximate boundary was drawn around the nucleus of Foxj1-negative cells based on the Hoechst signal. The mean gray intensity within this boundary was recorded as the p-rpS6 signal for that cell. For measuring Foxj1 intensity, a bioinformatic tool was used to restrict the nucleus outline using its Hoechst signal and then measure the mean gray intensity of Foxj1 within this area.

## Centriole number and patch area assessment

The number of centrioles in ependymal cells was counted manually from Z-stack images. ImageJ's "Multi-point" tool was used to mark centrioles and avoid errors. ImageJ "Polygon selection" tool calculates the patch area multiple centrioles occupy. Centrioles on the outer edge of the patch were used as pivot points to outline the patch. ImageJ will then calculate the enclosure area, considered the centriolar patch area.

## Time-lapse analysis

Ependymal cell cultures derived from Centrin2-GFP mice were used for this experiment to visualize centrioles in live cells. Images were acquired at 30-min intervals over a total duration of 48 h. After the acquisition, Z-projections of each frame were created in ImageJ. Only mononucleated cells were included in the analysis. A cell with only a pair of centrioles was identified as a progenitor cell. The appearance of a centriolar cloud and deuterosomes near the centrosomal centrioles marked the beginning of the A-phase. The onset of the G-phase was indicated by individual distinct centrioles on deuterosomes arranged in a pentamerous-like state. The first sign of individual centriole disengagement from deuterosomes marked the beginning of the D-phase. The D-phase ended when centrioles disengaged from the remaining deuterosome, indicating that the cell had become a fully differentiated ependymal. If centrioles failed to disengage from deuterosomes and the centriole-laden deuterosomes converged towards each other, it was termed deuterosome regrouping. Only cells that progressed from one differentiation phase to the next within the acquired movie frames were included in the analysis. The duration of time spent by cells in each phase was measured and used for further study.

## Immunoblot analysis

Cells and tissues were lysed in NP-40 lysis buffer (20 mM Tris pH 8.8; 138 mM NaCl; 5 mM EDTA; 2.7 mM KCl; 20 mM NaF; 5% glycerol; 1% NP-40- IGEPAL®; #CA-630; Sigma-Aldrich) supplemented with complete protease inhibitor and phosSTOP phosphatase inhibitor cocktails (Roche). Protein extracts were resolved by SDS-PAGE, transferred to PVDF membranes and incubated with the primary antibodies. The membranes were blocked in 5% nonfat dry milk in TBS containing 0.1% Tween 20 (TBST) for 1 h at room temperature. Primary antibodies were diluted in 5% bovine serum albumin (Sigma-Aldrich®) in TBST, and the membranes were incubated with diluted primary antibodies including cdc2 p34 mouse IgG2a kappa monoclonal (1:1000, Santa Cruz Biotechnology®, catalog: sc-54), Phospho-cdc2 (Tyr15) Rabbit polyclonal (1:1000, Cell Signaling Technology®, catalog: 9111), wee1 rabbit IgG monoclonal (1:1000, Cell Signaling Technology®, catalog:13084), Phospho-Wee1 (Ser642) rabbit IgG monoclonal (1:1000, Cell Signaling Technology®, catalog: 4910), rb rabbit IgG polyclonal (1:100, Santa Cruz Biotechnology®, catalog: sc-50), Phospho-Rb (Ser-807/811) rabbit polyclonal (1:1000, Cell Signaling Technology®, catalog: 9308), cyclin A rabbit IgG polyclonal (1:1000, Santa Cruz Biotechnology®, catalog: sc-596), cyclin D1 mouse IgG1 monoclonal (1:1000, Santa Cruz Biotechnology®, catalog: sc-450), p21 mouse monoclonal IgG2b (1:500, Santa Cruz Biotechnology®, catalog: sc-6246), p53 rabbit IgG polyclonal (1:500; Santa Cruz Biotechnology®, catalog: sc-6243), β-actin mouse IgG2b monoclonal (1:5000, Protein-tech®, catalog: 66009-1-Ig), α-tubulin mouse IgG1 monoclonal (1:2000, Sigma Aldrich®, catalog: T9026), GAPDH rabbit monoclonal (1:1000, Cell Signaling Technology®, catalog: 2118), cyclin O rabbit IgG polyclonal (1:500, Invitrogen®, catalog: PA5-20263), Myc-Tag mouse IgG2a kappa monoclonal (1:1000, Cell Signaling Technology®, catalog: 2276), CAD rabbit IgG monoclonal (1:1000, Cell Signaling Technology®, catalog: 93925), Phospho-CAD (Ser1859) rabbit polyclonal (1:1000, Cell Signaling Technology®, catalog: 12662), S6K1 rabbit IgG monoclonal (1:1000, Cell Signaling Technology®, catalog: 2708), phospho-S6K1

(Thr389) polyclonal (1:1000, Cell Signaling Technology®, catalog: 9205), S6 mouse IgG1 monoclonal (1:1000, Cell Signaling Technology®, catalog: 2317), phospho-S6 (Ser240/244) rabbit IgG monoclonal (1:1000, Cell Signaling Technology®, catalog: 5364), Akt rabbit polyclonal (1:1000, Cell Signaling Technology®, catalog: 9272), phospho-Akt rabbit IgG monoclonal (1:1000, Cell Signaling Technology®, catalog: 4060), phospho-GSKα/β (Ser 21/9) rabbit polyclonal (1:1000; Cell Signaling Technology®, catalog: 9331), Anti-GAS2L1 rabbit IgG polyclonal (1:1000; Sigma®, catalog: HPA019858, lot: 000056867) overnight at 4 °C. The next day, the membranes were washed three times with TBST for each 10 min at room temperature. After washing, they were incubated with diluted secondary antibodies including HRP-linked anti-rabbit (7074, Cell Signaling Technology®) and HRP-linked anti-mouse (7076, Cell Signaling Technology®) for 1 h at room temperature. After washing three times with TBST for each 10 min at room temperature, immunodetection was performed using ECL reaction reagents.

### Phosphoproteome enrichment and mass spectrometry

Cell lysates were prepared according to specifications from Cell Signaling Technology for phosphoproteomic analysis. Briefly, $2 \times 10^8$ cells of each condition were starved for 2 h in Earl's Balanced Salt Solution (EBSS; #24010043; Gibco) and then rinsed with ice-cold PBS. After carefully removing all PBS, cells were serially lysed in 10 ml of fresh urea lysis buffer (20 mM HEPES pH 8.0, 9 M urea, 1 mM activated sodium orthovanadate, 2.5 mM sodium pyrophosphate, 1 mM β-glycerol-phosphate) and passed through 21 G sting ten times, to produce ~20–40 mg of protein per condition. Lysates were frozen at −80 °C until processed for PTMScan analysis as previously described (Stokes et al, 2015) (Cell Signaling Technology).

Cellular extracts were sonicated, centrifuged, reduced with DTT, and alkylated with iodoacetamide. In all, 500 μg total protein for each sample was digested with trypsin and purified over C18 columns for enrichment with the Fe-IMAC (#20432) column. Enriched peptides were purified over C18 STAGE tips (Rappsilber et al, 2003). Enriched peptides were subjected to secondary digest with trypsin and a second STAGE tip before LC-MS/MS analysis.

Replicate injections of each sample were run non-sequentially for each enrichment. Peptides were eluted using a 90-min or 150-min linear acetonitrile gradient in 0.125% formic acid delivered at 280 nL/min. Tandem mass spectra were collected in a data-dependent manner with a Thermo Orbitrap QExactive or Fusion™ Lumos™ Tribrid™ mass spectrometer using a top-twenty MS/MS method, a dynamic repeat count of one, and a repeat duration of 30 s. Real-time recalibration of mass error was performed using lock mass (Olsen et al, 2005) with a singly charged polysiloxane ion $m/z = 371.101237$.

MS/MS spectra were evaluated using SEQUEST and the Core platform from Harvard University (Eng et al, 1994; Huttlin et al, 2010; Villen et al, 2007). Files were searched against the most recent update of the UniProt *mus musculus* FASTA database. A mass accuracy of $+/-5$ ppm was used for precursor ions and 0.02 Da for product ions. Enzyme specificity was limited to trypsin, with at least one tryptic (K- or R-containing) terminus required per peptide and up to four mis-cleavages allowed. Cysteine carboxamidomethylation was specified as a static modification, oxidation of methionine, and phosphorylation on serine,

threonine, or tyrosine residues were allowed as variable modifications. Reverse decoy databases were included for all searches to estimate false discovery rates, and filtered using a 2.5% FDR in the Linear Discriminant module of Core. Peptides were also manually filtered using a $-/+$ 5-ppm mass error range and the presence of a phosphorylated residue. All quantitative results were generated using Skyline (MacLean) to extract the integrated peak area of the corresponding peptide assignments in the MS1 channel. The accuracy of quantitative data was ensured by manual review in Skyline or in the ion chromatogram files.

### Statistical analysis

Statistical tests were done using GraphPad Prism 7 software version. The tests performed were unpaired, nonparametric, Mann–Whitney *t* test, or unpaired, nonparametric, Sidak's multiple comparison ANOVA test. For correlation assessment, Pearson's correlation coefficient was measured. Graphs were plotted with either GraphPad Prism 7 or with Microsoft Excel. *P* value associated with asterisk marks were ns: >0.05, *: ≤0.05 **: ≤0.05, ***: ≤0.01, ****: ≤0.001. Bars and error bars represent mean and standard deviation, respectively.

## Data availability

The mass spectrometry proteomics data have been deposited to the ProteomeXchange Consortium via the PRIDE partner repository with the dataset identifier PXD060982. Project Webpage: https://www.ebi.ac.uk/pride/archive/projects/PXD060982. FTP Download: https://ftp.pride.ebi.ac.uk/pride/data/archive/2025/03/PXD060982. The metabolomics data are included as CSV file in the presented Dataset, as suggested by *EMBO Reports* Author guidelines.

The source data of this paper are collected in the following database record: biostudies:S-SCDT-10_1038-S44319-025-00460-2.

## Peer review information

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

## Acknowledgements

We are grateful to the members of the Spassky and Pende labs for their support, critical reading of the manuscript, and helpful discussions. We thank the histology, imaging, metabolomics, viral vector facilities, neurobehaviour, animal facilities at the SFR Necker and Nathalie Servel of the radioactivity core facility of the INEM for the technical help. We thank the IBENS administrative team and imaging platform for their support and the IBENS Animal Facility for animal care and excellent support. This work was supported by the INSERM, the CNRS, the Ecole normale supérieure (ENS), and grants from the European Research Council (ERC consolidator grant 647466 to NS and 616917 to MP), the Fondation pour la Recherche Médicale (FRM, EQU 202103012767 to NS), MSD Avenir, RHU-Cosy, ARC and Inca to MP, the ANR (ANR-21-CE16-0016) to NS/MP, ANR-20-CE45-0019 to NS/AG, and ANR-22-CE16-0011 to NS. KT was supported by an ARC fellowship, a FRM fellowship supported AB and AS received a fellowship from the Labex MEMOLIFE.

## Author contributions

**Alexia Bankolé**: Investigation. **Ayush Srivastava**: Investigation. **Asm Shihavuddin**: Methodology. **Khaled Tighanimine**: Investigation. **Marion Faucourt**: Investigation. **Vonda Koka**: Investigation. **Solene Weill**: Investigation. **Ivan Nemazanyy**: Investigation. **Alissa J Nelson**: Investigation. **Matthew P Stokes**: Investigation. **Nathalie Delgehyr**: Investigation. **Auguste Genovesio**: Methodology. **Alice Meunier**: Supervision. **Stefano Fumagalli**: Supervision; Writing—original draft; Writing—review and editing. **Mario Pende**: Conceptualization; Supervision; Writing—original draft; Writing—review and editing. **Nathalie Spassky**: Conceptualization; Supervision; Funding acquisition; Validation; Investigation; Visualization; Writing—original draft; Writing—review and editing.

Source data underlying figure panels in this paper may have individual authorship assigned. Where available, figure panel/source data authorship is listed in the following database record: biostudies:S-SCDT-10_1038-S44319-025-00460-2.

## Disclosure and competing interests statement

The authors declare no competing interests.

# Expanded View Figures

**Figure EV1.  mTORC1 signaling pathway is active during centriole amplification.**

(**A, B**) P4 brain lateral ventricular en face, labeling Centrin (centrioles, white), p-rpS6 (red) and GT335 (cilia, green) or p21 (green) in (**A, B**), respectively. (**C**) Ependymal cells in respective phases of differentiation characterized by their staining of Centrin (centrioles, white) and GT335 (cilia, green), and p-rpS6 (red). It is worth noting that the expression of p-rpS6 is during the intermediate stages of differentiation. (**D**) Regression plot testing the correlation between apical area and centriole number in mature ependymal cells in Tsc1 cKO at P0, rapamycin injected pups at P4 and their respective controls, $R^2$ is the correlation coefficient; ns $P = 0.0542$; **$P = 0.0023$; ***$P = 0.0001$; ****$P < 0{,}0001$. Data information: In (**D**), **$P < 0.01$, ***$P < 0.001$, ****$P < 0.0001$ (Pearson's correlation test). $n$ = number of cells >50. Scale bars: 10 μm.

   

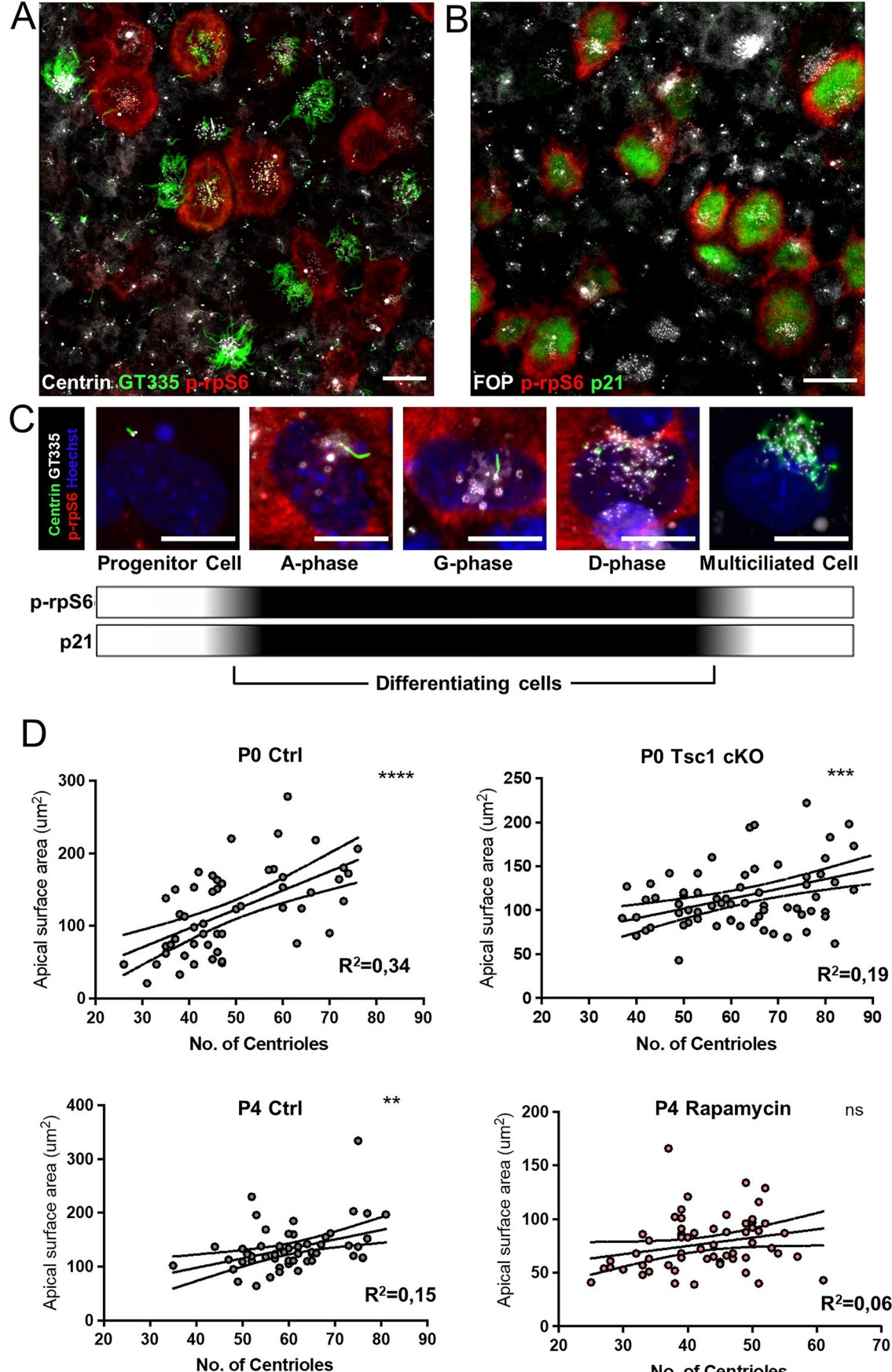

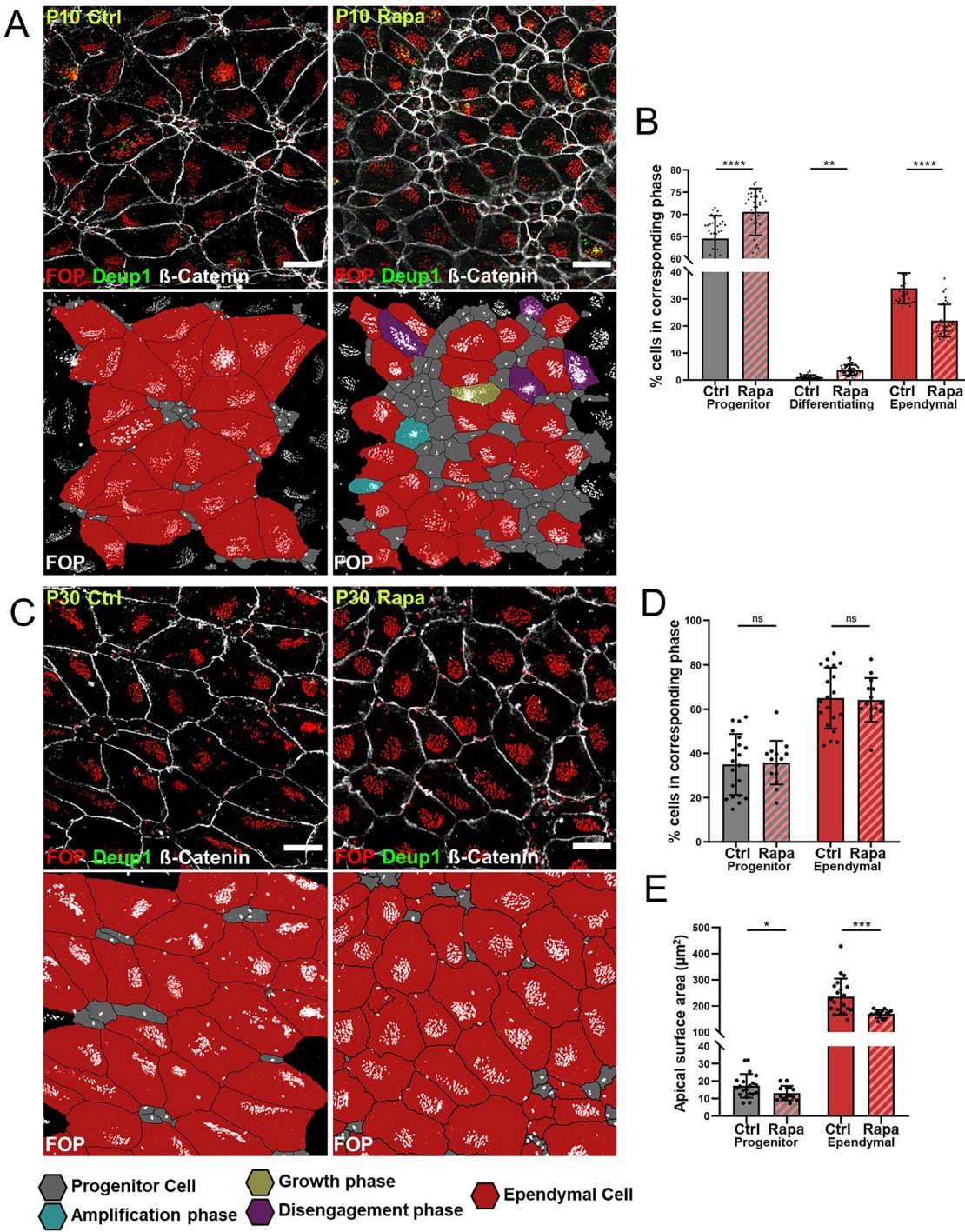

**Figure EV2. Long-term effect of rapamycin treatment.**

(A) Lateral ventricular wall of mice at P10 injected with rapamycin every day between P0 and P4 and labeled with Fop (centrioles, red), Deup (deuterosomes, green) and β-Catenin (cell junction, white). Corresponding segmented images are shown below. (B) Percentage of cells at each stage of differentiation at P10; **$P$ = 0.0022; ****$P$ < 0.0001. (C) Lateral ventricular wall of mice at P30 injected with rapamycin every day between P0 and P4 and labeled with Fop (centrioles, red), Deup (deuterosomes, green) and β-Catenin (cell junction, white). Corresponding segmented images are shown below. (D) Percentage of cells at each stage of differentiation at P30; ns $P$ = 0.9039. (E) Quantification of the size of the apical surface of progenitor and mature ependymal cells in rapamycin and control conditions at P30; * $P$ = 0.0470; ***$P$ = 0.0005. Scale bars: 8.5 μm. Data information: In (B, D, E), data are presented as mean ± SD. ****$P$ < 0.0001, ***$P$ < 0.001, **$P$ < 0.01, *$P$ ≤ 0.05, ns not significant (Student's $t$ test).

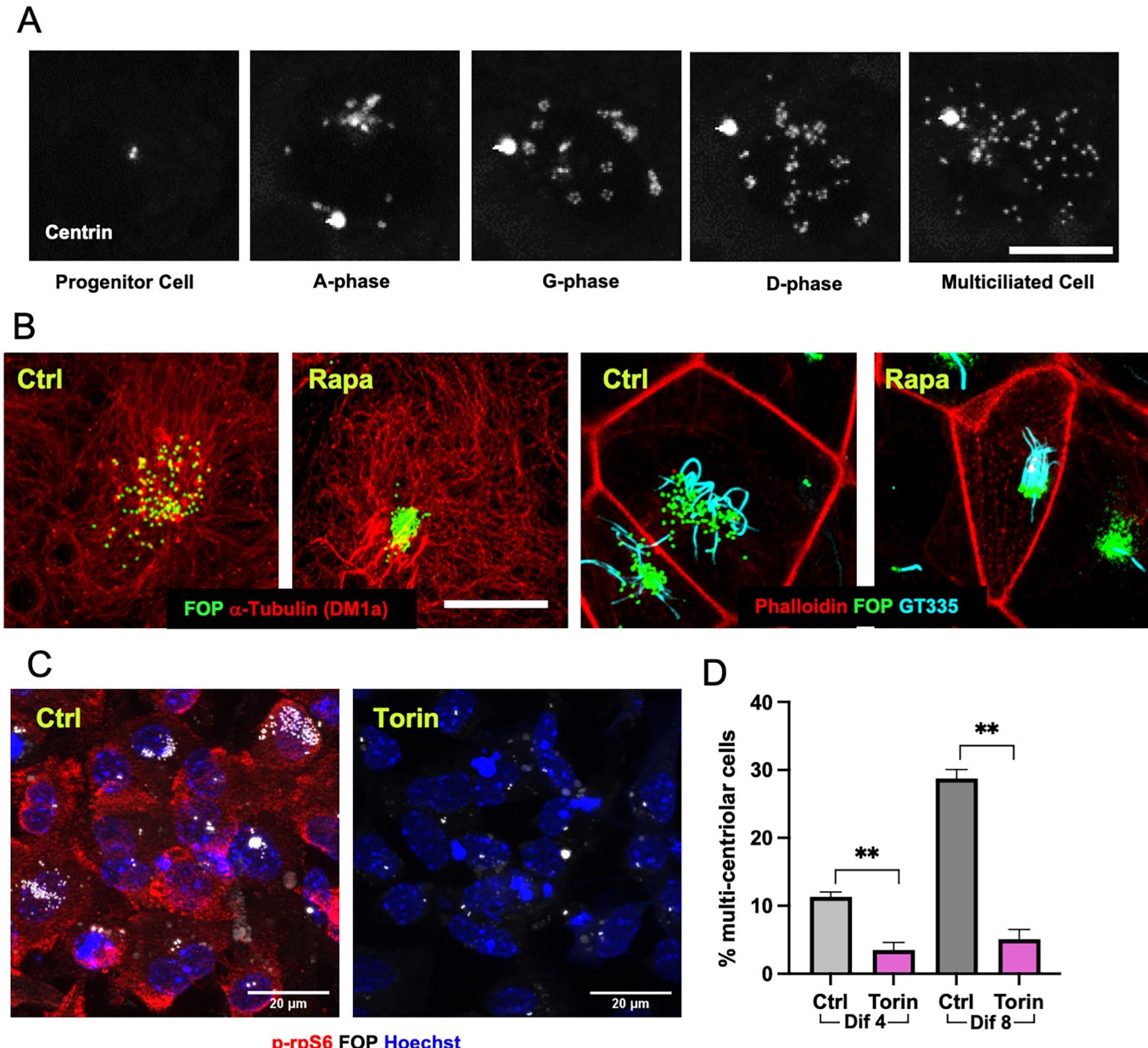

**Figure EV3.    In vitro stages of ependymal differentiation.**

(A) Representative images of Cen2GFP+ cells at different stages of basal body formation during ependymal cell differentiation in vitro. (B) Representative images of primary cells at div 4 immunolabeled with FOP and α-Tubulin (DM1a) or Phalloidin and GT335 in control and rapamycin conditions. (C) Representative images of primary cells at dif 4 in control and Torin conditions. Control is the same as in Fig. EV4D, as the experiments were carried out in parallel. (D) Percentage of multicentriolar cells at different time points in the indicated conditions; **$P$ = 0.0022. Data information: (D) presents data as mean ± SEM. **$P < 0.01$ (Mann–Whitney test); $n$ = 3. Scale bars: 10 µm (A, B), 20 µm (C).

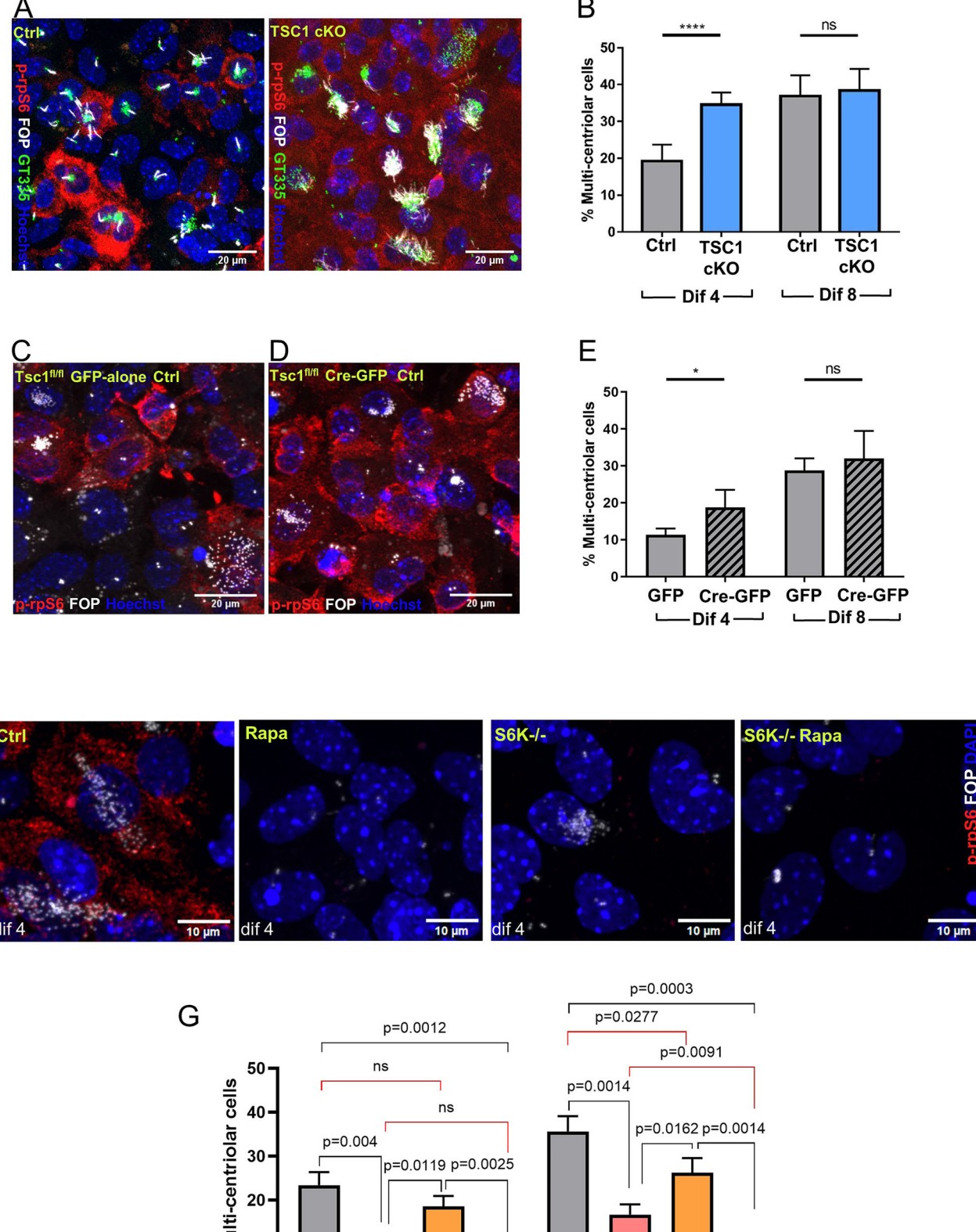

◄ **Figure EV4. Inactivation of Tsc1 in Nestin progenitor cells leads to increased ependymal cell differentiation, while S6K inactivation does not abolish rapamycin sensitivity.**

(A) Representative images of primary cells from control and Tsc1 cKO mice at div 4 immunolabeled with FOP (centrioles), GT335, and pS6 antibodies. (B) Percentage of multicentriolar cells at div 4 and div 8; ****$P < 0.0001$; ns $P = 0.8182$. (C, D) Representative images of primary cells from Tsc1fl/fl mice infected with GFP (C) or Cre-GFP Adenovirus. Cells are immunolabeled with FOP and pS6 antibodies. Control is the same as in Fig. EV3C, as the experiments were carried out in parallel. (E) Percentage of multicentriolar cells at div 4 and div 8 in all conditions; *$P = 0.043$; ns $P = 0.6991$. (F) Representative pictures of wild-type or S6K$^{-/-}$ primary cells at 4 days in vitro, immunolabelled with pS6 as a marker of mTORC1 and FOP to detect multibasal bodies (MBB) cells in differentiated cells. The top panels are cells treated with vehicle (EtOH), and the bottom panels are cells treated with 20 nM Rapamycin. The arrows show examples of MBB cells. div differentiation in vitro, WT wild type. (G) Fold changes of differentiated ependymal cell number by counting cells with multibasal bodies. The bars represent the mean $+/-$ SEM; each dot is one replicate, $n = 6$. $P$ values were determined using a two-sided Student's $t$ test. **$P < 0.01$ and ***$P < 0.001$. Data information: In (B, E), data are presented as mean ± SD. ****$P < 0.0001$, *$P \leq 0.05$, ns not significant (Student's $t$ test). In (G), the bars represent the mean $+/-$ SEM; each dot is one replicate, $n = 6$. $P$ values were determined using a two-sided Student's $t$ test. **$P < 0.01$ and ***$P < 0.001$. Ctrl Control.

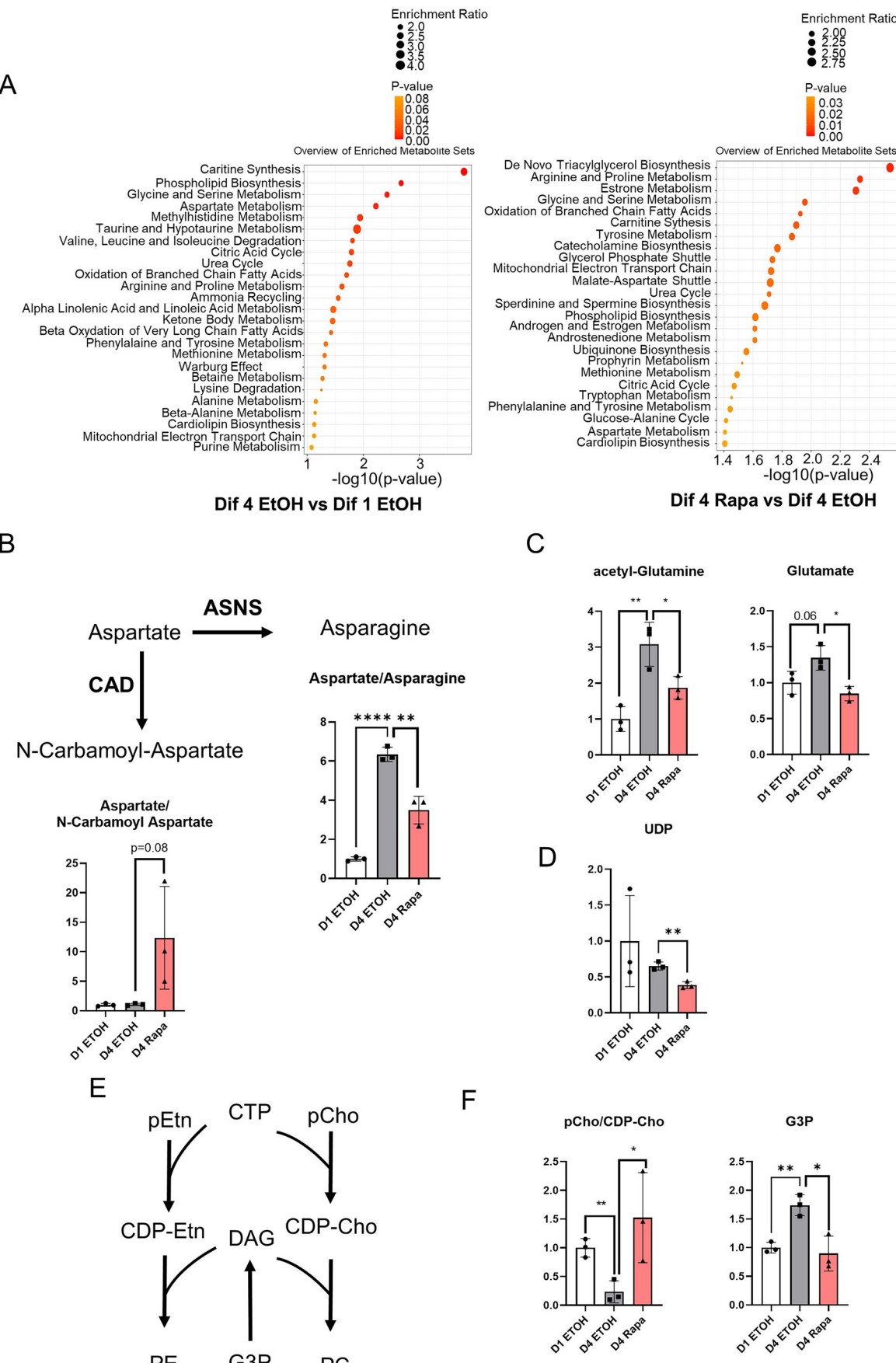

◄ **Figure EV5. Metabolic rewiring by mTORC1.**

(A) KEGG-based pathway enrichment analyses of the metabolomes of progenitors and ependymal cells (left panel), or ependymal cells treated or not with Rapamycin. (B) Summary of the metabolic profiling of the aspartate metabolism in progenitors and ependymal cells treated or not with Rapamycin. (C) Levels of the indicated metabolites involved in glutamine metabolism in progenitors and ependymal cells treated or not with Rapamycin. (D) Levels of UDP in progenitors and ependymal cells treated or not with Rapamycin. (E) Scheme depicting the phospholipids biosynthesis pathways and highlighting the requirement of CTP and DAG for the second and third steps, respectively. (F) Ratios of the indicated metabolites and levels of G3P in progenitors and ependymal cells treated or not with Rapamycin. Data information: For (B, C, D, F), the bars represent the mean $+/-$ SEM, each dot is one replicate, $n = 3$. Data are presented as mean $\pm$ SD. ****$P < 0.0001$, ***$P < 0.001$, **$P < 0.01$, *$P \leq 0.05$ (Student's $t$ test).

