## [Peer Review File · EMBO Reports]

mTOR controls ependymal differentiation by targeting alternative cell cycle and centrosomal proteins

Mario Pende, Alexia Bankolé, Ayush Srivastava, Asm Shihavuddin, Khaled Tighanimine, Marion Faucourt, Vonda Koka, Solene Weill, Ivan Nemazanyy, Alissa Nelson, Matthew Stokes, Nathalie Delgehr, Auguste Genovesio, Alice Meunier, Stefano Fumagalli, and Nathalie Spassky

Corresponding author(s): *Nathalie Spassky (nathalie.spassky@bio.ens.psl.eu)* , *Mario Pende (mario.pende@inserm.fr)*

Review Timeline:

Submission Date:	21st Oct 24
Editorial Decision:	9th Dec 24
Revision Received:	13th Feb 25
Editorial Decision:	12th Mar 25
Revision Received:	18th Mar 25
Accepted:	4th Apr 25

Editor: Deniz Senyilmaz Tiebe

Transaction Report:

Dear Dr. Pende,

Thank you for transferring your research manuscript to our journal, which was now seen by three referees, whose reports are copied below.

I apologize for this unusual delay in getting back to you. It took longer than anticipated to receive the referee reports.

Referees express interest in the proposed mechanism by which mTOR regulates ependymal differentiation. However, they also raise some concerns that need to be addressed to consider publication here.

I find the reports informed and constructive, and believe that addressing the concerns raised will significantly strengthen the manuscript. As the reports are below, and I think all points need to be addressed, I will not detail them here. Please contact me if you have questions or comments regarding the revision for further discussion (also by video chat).

Given these positive recommendations, we would like to invite you to submit a revised manuscript. Please revise your manuscript with the understanding that the referee concerns (as in their reports) must be fully addressed and their suggestions taken on board. Please address all referee concerns in a complete point-by-point response. Acceptance of the manuscript will depend on a positive outcome of a second round of review. It is EMBO reports policy to allow a single round of major experimental revision only and acceptance or rejection of the manuscript will therefore depend on the completeness of your responses included in the next, final version of the manuscript.

We realize that it is difficult to revise to a specific deadline. In the interest of protecting the conceptual advance provided by the work, we recommend a revision within 3 months. Please discuss the revision progress ahead of this time with me if you require more time to complete the revisions, or if you have questions or comments regarding the revision (also by video chat).

1. A data availability section providing access to data deposited in public databases is missing (where applicable).
2. Your manuscript contains statistics and error bars based on $n=2$. Please use scatter plots in these cases.

You can submit the revision either as a Scientific Report or as a Research Article. For Scientific Reports, the revised manuscript can contain up to 5 main figures and 5 Expanded View figures, and it should not exceed 27000 characters. If the revision leads to a manuscript with more than 5 main figures it will be published as a Research Article. In this case the Results and Discussion section should be separate. If a Scientific Report is submitted, these sections have to be combined. This will help to shorten the manuscript text by eliminating some redundancy that is inevitable when discussing the same experiments twice. In either case, all materials and methods should be included in the main manuscript file.

3) We replaced Supplementary Information with Expanded View (EV) Figures and Tables that are collapsible/expandable online. A maximum of 5 EV Figures can be typeset. EV Figures should be cited as 'Figure EV1, Figure EV2' etc... in the text and their respective legends should be included in the main text after the legends of regular figures.

4) a .docx formatted letter INCLUDING the reviewers' reports and your detailed point-by-point responses to their comments. As

part of the EMBO publication's Transparent Editorial Process, EMBO reports publishes online a Review Process File (RPF) to accompany accepted manuscripts. This File will be published in conjunction with your paper and will include the referee reports, your point-by-point response and all pertinent correspondence relating to the manuscript.

<https://www.embopress.org/page/journal/14693178/authorguide#transparentprocess>

5) a complete author checklist, which you can download from our author guidelines

<https://www.embopress.org/page/journal/14693178/authorguide>. Please insert information in the checklist that is also reflected in the manuscript. The completed author checklist will also be part of the RPF.

6) Please note that all corresponding authors are required to supply an ORCID ID for their name upon submission of a revised manuscript (<<https://orcid.org/>>). Please find instructions on how to link your ORCID ID to your account in our manuscript tracking system in our Author guidelines

<<https://www.embopress.org/page/journal/14693178/authorguide#authorshipguidelines>>

7) Before submitting your revision, primary datasets produced in this study need to be deposited in an appropriate public database (see <https://www.embopress.org/page/journal/14693178/authorguide#datadeposition>). Please remember to provide a reviewer password if the datasets are not yet public. The accession numbers and database should be listed in a formal "Data Availability" section placed after Materials & Method (see also

<https://www.embopress.org/page/journal/14693178/authorguide#datadeposition>). Please note that the Data Availability Section is restricted to new primary data that are part of this study. * Note - All links should resolve to a page where the data can be accessed. *

Additional information on source data and instruction on how to label the files are available:

<https://www.embopress.org/page/journal/14693178/authorguide#sourcedata>

9) Our journal encourages inclusion of *data citations in the reference list* to directly cite datasets that were re-used and obtained from public databases. Data citations in the article text are distinct from normal bibliographical citations and should directly link to the database records from which the data can be accessed. In the main text, data citations are formatted as follows: "Data ref: Smith et al, 2001" or "Data ref: NCBI Sequence Read Archive PRJNA342805, 2017". In the Reference list, data citations must be labeled with "[DATASET]". A data reference must provide the database name, accession number/identifiers and a resolvable link to the landing page from which the data can be accessed at the end of the reference. Further instructions are available at <http://www.embopress.org/page/journal/14693178/authorguide#referencesformat>

10) Regarding data quantification (see Figure Legends:

<https://www.embopress.org/page/journal/14693178/authorguide#figureformat>)

11) The journal requires a statement specifying whether or not authors have competing interests (defined as all potential or

actual interests that could be perceived to influence the presentation or interpretation of an article). In case of competing interests, this must be specified in your disclosure statement. Further information: <https://www.embopress.org/competing-interests>

12) Please also note our reference format:

13) All Materials and Methods need to be described in the main text using our 'Structured Methods' format, which is required for all research articles. According to this format, the Methods section includes a Reagents and Tools Table (listing key reagents, experimental models, software and relevant equipment and including their sources and relevant identifiers) followed by a Methods and Protocols section describing the methods using a step-by-step protocol format. The aim is to facilitate adoption of the methodologies across labs. More information on how to adhere to this format as well as a downloadable template (.docx) for the Reagents and Tools Table can be found in our author guidelines:

I look forward to seeing a revised version of your manuscript when it is ready. Please let me know if you have questions or comments regarding the revision.

Kind regards,

Deniz Senyilmaz Tiebe

Deniz Senyilmaz Tiebe, PhD
Senior Scientific Editor
EMBO Reports

Referee #1:

The manuscript by Bankole and colleagues is focused on characterizing the role(s) of mTORC1 pathway by pharmacological and genetic interventions to investigate the impact on ependymal cell cycle and differentiation. The authors demonstrate that mTORC1 inhibition (by rapamycin) promotes a quiescent cell state, dampening the expression of cell cycle proteins required for centriole amplification. They also show the epistatic interaction with master regulators of ependymal cell differentiation (Mcidas/E2F4) and with one of the downstream effectors of mTOR1 (S6 Kinase). Subsequently, they analyzed the metabolome and the phosphoproteome during ependymal cell differentiation in the presence or absence of mTORC1 activity, and find multiple targets of mTORC1 with enrichment in proteins involved in the cell cycle and cytoskeletal dynamics. Specifically, they show the actin and microtubule cytolinker Gas2L1 protein controls centriolar number and density downstream of the mTORC1 pathway.

Overall, this is an interesting study, the work is well performed, and the data are convincing and well presented - as is usual from this group. Therefore, I believe the study should be published with a few minor revisions.

Main comments:

1) Fig.2 and page 8-9 - Via rapamycin treatment and TSC cKO, the authors suggest that "mTOR controls the timing of centriole amplification, the increase in apical contact size, and the number of centrioles". Indeed, it looks like modulating mTORC1 activity can change centriole number as well as surface area size of each cell (Fig 2D-K). But does the ratio of centriole number and the surface area per cell actually change? If the authors plotted centriole number relative to surface area size for each cell, would there be a change in the relationship between centriole number and surface area/centriole patch per ependymal cell, or will that ratio (slope of the line) remain consistent upon mTORC1 modulation? Because one possibility is that you simply get smaller cells with less centrioles (or larger cells with more centrioles), but the same density/unit area of the cell surface. Distinguishing between these two possibilities will be important.

2) Along these lines, the authors discuss the relationship between surface area, centriole patch size and centriole number in several instances throughout the manuscript, and they cite the study by Kulkarni et al (2021) as supportive data. However, the relationship between centriole number scaling and surface area in mammalian multiciliated cells were described in detail by

Nanjundappa et al (eLife, 2019) and LoMastro et al (eLife, 2022). I think it is important for the authors to cite these papers and put them in context, since the experiments in the Kulkarni study were performed in the *Xenopus* model.

3) Fig 3H-I: rapamycin treatment resulted in deuterosomes regrouping and impaired centriole ability to disperse from their platforms. This is a very interesting result, yet it is not clear how this regrouping occurs once mTORC1 is inhibited. Are there any changes to cell size during the rapa treatment? How about effects on the actin and microtubule cytoskeleton, which play important roles in centriole migration, organization and spacing? Could mTORC1 modulation be affecting the cytoskeleton, and thereby indirectly affecting centriole number and spacing?

4) Fig 6: They demonstrate that a 25 hour treatment with rapamycin during ependymal differentiation was sufficient to reduce the expression of G1 markers (P-Rb, Cyclin D1), G2 markers (Cyclin A, P-Cdk1, Wee1), replication stress markers (p21), and ependymal cell-specific cyclins (Cyclin O). Based on this, they conclude that "... mTORC1 inhibition blunts ependymal cell differentiation, at least in part, by acting on the proteins of the alternative cell cycle for centriole amplification." Do the authors think that mTORC1 activity is acting separately on these cell cycle proteins for each of those centriole transition stages in ependymal cells? And how could they separate its function(s) during early differentiation versus during these stages of centriole amplification, growth and disengagement? In Fig 7, they then show that mTORC1 acts upstream of Mcidas/E2F4, probably at the late G1 phase of the alternative cell cycle in differentiating cells. Therefore, the fact that G2 and other late markers are downregulated after 25 hour treatment with rapamycin are unlikely to be direct effects on those proteins.

5) Fig 9: They show that overexpression of the phosphor-mutant Gas2L1 does not impact the number of multicentriolar cells. Is it possible that the wildtype copy of the gene may be affecting the experiment? Have the authors tried overexpressing them in Gas2L1-depleted cells?

6) Minor point: the font sizes on some of the figure panels are too small are really difficult to read (e.g. Fig 4A-B, 8A, Supp 5A)

Referee #2:

The manuscript from Bankole' et al is interesting and overall well written, it contains remarkable in vivo and in vitro data and novel techniques. I have few comments which should be addressed in my opinion to strengthen the manuscript and clarify some of the claims. I overall consider this manuscript worth being published by EMBO reports.

Here are my comments:

1- In the introduction Authors could consider adding a paragraph regarding the function of ependymal cells since this is not a neuro focused journal and some of the readers might not be familiar with the cell type.

2- mTORC1 inactivation is only done by the use of Rapamycin, some of the in vitro experiments in fig 3 and 6 should be repeated using for example torin1. Rapamycin is known to not affect TFEB/TFE3 phosphorylation in wildtype cells whereas Torin1 pushes TFEB/TFE3 heavily into the nucleus blunting mTORC1 phosphorylation. If the effect of Torin1 in your experiments is not similar to Rapamycin, this could suggest a role of TFEB/TFE3 in the process.

3- Consider moving Fig 5 to the supplementary figures to shorten manuscript

4- In figure 7 MCIDAS and E2F4 are overexpressed together, in my opinion there is a lack of controls (overexpression of the 2 alone), at least the WB in figure 7a should be repeated with the single overexpression.

5- In the same figure the Authors claim that overexpressing MCIDAS and E2FA increases mTORC1 activity, quantification in 7b shows that only pS6 is increased statistically, whereas pS6K and pCAD are not.

This claim of mTORC1 increased activation is fascinating but should be strengthen for example by checking also p4EBP1 or in alternative since it is not a key part of the actual story could be reconsidered for future work, it feels also a bit too speculative in the discussion. The single overexpression of MCIDAS and E2F4 could also help clarify if either of them can activate mTORC1 on its own.

Very minor comments:

1- there is a reference of figure 5D which is not present, I imagine the authors meant 4D.

2- In the phosphoproteomic screening chapter Authors wrote: out of 57 mTORC1 substrates reported in a recent review etc, without citing the review, this should be added.

Referee #3:

In this manuscript, Bakolé, Srivastava and colleagues perform an extensive body of work to uncover new roles of mTOR signalling across ependymal cell differentiation. Using gain and loss of function studies in vivo and in vitro and a range of technological approaches including metabolic profiling and phosphoproteomics, the authors demonstrate that mTORC1 is necessary for cell growth, metabolic rewiring, centriole amplification and disengagement during ependymal cell differentiation. The authors go on to show that inhibition of mTORC1 blocks the expression of cell cycle genes involved in the multiciliation cell cycle and centriole amplification, blocking ependymal cell differentiation. mTORC1 acts upstream of Mcidas, a master regulator

of ependymal cell differentiation, and Mcidas can, in turn, activate mTORC1 during ependymal cell differentiation. Using phosphoproteomic and photo-mutant analysis, the authors discover new mTORC1 targets during ependymal cell differentiation and show that mTORC1-mediated phosphorylation of Gas2L1 drives centriole disengagement.

The work is convincing and the phenotypes presented are strong. The study is also timely as it builds upon recent discoveries, including the role of mTOR in regulating the size of cells lining the brain lateral ventricles and the multiciliation cell cycle, and will be of great interest to the cell and molecular biology communities as well as developmental biologists interested in brain development and developmental disorders such as hydrocephalus.

There are some minor issues that the authors should address before publication in EMBO reports.

1. The manuscript is well written, but the amount of information and 9 figures makes it challenging to read. To highlight the main points of the story, I recommend shortening the text and reducing the number of figures by 1-2. For example, the data presented in Figure 1 is important, but it is not novel so it could be made a supplementary figure. Figure 5 does not add to the main points of the story so it could also be made a supplementary figure and the text related to it shortened. Adding a summary figure will be great to bring together the many and important findings of the study.

2. Figure 1D-F. Based on what is plotted, each dot in Figure 1D and F probably represents one mouse (one wall of the lateral ventricle). Does each dot in Figure 1F represent one cell? Please add this information in the figure legend.

3. Figure 1G. 26% and 67% of the cells lining the lateral walls are classified as ependymal cells in the P7-P10 and P15-P45 groups, respectively. However, most ependymal cells in the brain are thought to complete their differentiation into multiciliated cells during the first postnatal week (Spassky et al. 2015). Can the authors comment on this? Is it possible that adult neural stem cells are classified as progenitor cells in this analysis? I guess one way of answering this is, does the number of FOXJ1-positive cells in the lateral wall/regions analysed differ between control and rapamycin-treated pups as the authors later find in vitro?

4. Figure 2. What do dots represent in each plot? How many samples were analysed for each experiment and experimental group? Please include this information in the figure legend.

5. Figure legend for Figure 2. Please correct a typo in the scale bar units.

6. In page 10, the authors take advantage of in vitro ependymal cell cultures to study how mTORC1 regulates ependymal cell differentiation. Among others, they find that inhibiting mTOR signalling by rapamycin results in fewer FOXJ1-positive cells but activation of mTOR signalling in Tsc1 cKO cells accelerates centriole amplification but does not result in more multiciliated (potentially FOXJ1-positive) cells. From this, the authors conclude that "(...) mTORC1 activation triggers ependymal cell differentiation but not its specification". Because there are fewer FOXJ1-positive cells in rapamycin-treated cultures than in controls, would a more accurate conclusion be that mTOR signalling is necessary but not sufficient for ependymal cell specification?

7. To link mechanical forces (related to brain growth) and mTOR signalling, an interesting experiment would be to wash out rapamycin from cultures after 4 days and allow the cells to recover. Does the number of multiciliated cells recover as in in vivo experiments? This experiment is not required for publication, but will be interesting to know the answer.

8. In page 12 "mostly due to reduced N-carbamoyl-aspartate levels (Fig. 5D)" should be Fig. 4D.

9. In page 16, "Out of 57 mTORC1 substrates reported in a recent review in different cell types, 14 were significantly down-regulated in rapamycin-treated ependymal cells, highlighting the ability to find target phosphorylation sites of interest using this methodology." Please add reference.

10. Figure 1: include scale bars in all microscopy images.

11. Mouse proteins (and antibodies) should be all capitals, please revise this throughout the text and figures.

12. The referencing style may need amending to match EMBO reports' style. Currently, the first two authors are listed in in-text references, whether they are co-first authors or not (only first or co-first authors should appear in in-text references).

Referee #1:

The manuscript by Bankole and colleagues is focused on characterizing the role(s) of mTORC1 pathway by pharmacological and genetic interventions to investigate the impact on ependymal cell cycle and differentiation. The authors demonstrate that mTORC1 inhibition (by rapamycin) promotes a quiescent cell state, dampening the expression of cell cycle proteins required for centriole amplification. They also show the epistatic interaction with master regulators of ependymal cell differentiation (Mcidas/E2F4) and with one of the downstream effectors of mTOR1 (S6 Kinase). Subsequently, they analyzed the metabolome and the phosphoproteome during ependymal cell differentiation in the presence or absence of mTORC1 activity, and find multiple targets of mTORC1 with enrichment in proteins involved in the cell cycle and cytoskeletal dynamics. Specifically, they show the actin and microtubule cytolinker Gas2L1 protein controls centriolar number and density downstream of the mTORC1 pathway.

Overall, this is an interesting study, the work is well performed, and the data are convincing and well presented - as is usual from this group. Therefore, I believe the study should be published with a few minor revisions.

Main comments:

1) Fig.2 and page 8-9 - Via rapamycin treatment and TSC cKO, the authors suggest that "mTOR controls the timing of centriole amplification, the increase in apical contact size, and the number of centrioles". Indeed, it looks like modulating mTORC1 activity can change centriole number as well as surface area size of each cell (Fig 2D-K). But does the ratio of centriole number and the surface area per cell actually change? If the authors plotted centriole number relative to surface area size for each cell, would there be a change in the relationship between centriole number and surface area/centriole patch per ependymal cell, or will that ratio (slope of the line) remain consistent upon mTORC1 modulation? Because one possibility is that you simply get smaller cells with less centrioles (or larger cells with more centrioles), but the same density/unit area of the cell surface. Distinguishing between these two possibilities will be important.

The relationship between centriole number and surface area in multiciliated ependymal cells among controls at birth is weak ($R^2=0.34$) and tends to decrease at P4 ($R^2=0.15$). However, the slope of the line is statistically significant, indicating that this correlation exists, although other factors may also contribute to regulating ependymal cell size. Interestingly, mTORC1 modulation results in a lower correlation at birth (TSC cKO: $R^2=0.19$) or even a complete loss of correlation

at P4 (Rapamycin: $R^2=0.06$), suggesting that changes in cell size alone cannot explain the phenotypes observed in our study. These results are presented in Expanded View EV 1D.

2) Along these lines, the authors discuss the relationship between surface area, centriole patch size and centriole number in several instances throughout the manuscript, and they cite the study by Kulkarni et al (2021) as supportive data. However, the relationship between centriole number scaling and surface area in mammalian multiciliated cells were described in detail by Nanjundappa et al (eLife, 2019) and LoMastro et al (eLife, 2022). I think it is important for the authors to cite these papers and put them in context, since the experiments in the Kulkarni study were performed in the *Xenopus* model.

We agree with the reviewer's comment and modified the text accordingly, citing the relevant literature.

3) Fig 3H-I: rapamycin treatment resulted in deuterosomes regrouping and impaired centriole ability to disperse from their platforms. This is a very interesting result, yet it is not clear how this regrouping occurs once mTORC1 is inhibited. Are there any changes to cell size during the rapa treatment? How about effects on the actin and microtubule cytoskeleton, which play important roles in centriole migration, organization and spacing? Could mTORC1 modulation be affecting the cytoskeleton, and thereby indirectly affecting centriole number and spacing?

An implication of cell size is intriguing. On the other hand, the size of the cells in the culture system is heterogeneous, independently of drug treatment, and it is therefore impossible to draw any conclusion about rapamycin-induced changes on cell size *in vitro*. We performed acetylated α -tubulin or phalloidin staining on rapamycin-treated and control ependymal cultures to examine the microtubule and actin cytoskeletons, respectively. We did not observe significant changes in their density and organization in rapamycin-treated cells compared to controls, suggesting that centriole regrouping is not due to cytoskeleton defects (Figure Expanded View EV3B).

4) Fig 6: They demonstrate that a 25 hour treatment with rapamycin during ependymal differentiation was sufficient to reduce the expression of G1 markers (P-Rb, Cyclin D1), G2 markers (Cyclin A, P-Cdk1, Wee1), replication stress markers (p21), and ependymal cell-specific cyclins (Cyclin O). Based on this, they conclude that "... mTORC1 inhibition blunts ependymal cell differentiation, at least in part, by acting on the proteins of the alternative cell cycle for centriole amplification." Do the authors think that mTORC1

activity is acting separately on these cell cycle proteins for each of those centriole transition stages in ependymal cells? And how could they separate its function(s) during early differentiation versus during these stages of centriole amplification, growth and disengagement? In Fig 7, they then show that mTORC1 acts upstream of Mcidas/E2F4, probably at the late G1 phase of the alternative cell cycle in differentiating cells. Therefore, the fact that G2 and other late markers are downregulated after 25 hour treatment with rapamycin are unlikely to be direct effects on those proteins.

We agree with the reviewer, and in this revised version, we clearly state in the text that the simplest and most likely explanation is that mTORC1 acts on G1 phase markers and the following changes in G2 markers are a consequence of this primary G1 effect. The model in Figure 8G clarifies this possibility.

5) Fig 9: They show that overexpression of the phosphor-mutant Gas2L1 does not impact the number of multicentriolar cells. Is it possible that the wildtype copy of the gene may be affecting the experiment? Have the authors tried overexpressing them in Gas2L1-depleted cells?

We used commercially available Adenoviral vectors from Vector Biolabs and the Gas2L1 cDNA was not designed to be resistant to shRNA. Moreover, the requested experiment would double the amount virus to be transduced and may decrease cell viability in these primary cultures.

6) Minor point: the font sizes on some of the figure panels are too small are really difficult to read (e.g. Fig 4A-B, 8A, Supp 5A)

We increased the font sizes.

Referee #2:

The manuscript from Bankole' et al is interesting and overall well written, it contains remarkable in vivo and in vitro data and novel techniques. I have few comments which should be addressed in my opinion to strengthen the manuscript and clarify some of the claims. I overall consider this manuscript worth being published by EMBO reports.

Here are my comments:

1- In the introduction Authors could consider adding a paragraph regarding the function of ependymal cells since this is not a neuro focused journal and some of the readers might not be familiar with the cell type.

The introduction has been updated accordingly. The first paragraph introduces the ependymal cells.

2- mTORC1 inactivation is only done by the use of Rapamycin, some of the in vitro experiments in fig 3 and 6 should be repeated using for example torin1. Rapamycin is known to not affect TFEB/TFE3 phosphorylation in wildtype cells whereas Torin1 pushes TFEB/TFE3 heavily into the nucleus blunting mTORC1 phosphorylation. If the effect of Torin1 in your experiments is not similar to Rapamycin, this could suggest a role of TFEB/TFE3 in the process.

As suggested, we assessed an additional drug in the mTOR pathway, Torin 1, on cultured ependymal cells. Treatment with Torin 1 decreased p-rpS6 expression and the number of multiciliated cells at Dif4 and Dif8, reflecting the effects of rapamycin. These results suggest that the canonical mTORC1 pathway is involved in ependymal cell differentiation and have been included as a Supplementary Figure (Supplementary Figure 3C).

3- Consider moving Fig 5 to the supplementary figures to shorten manuscript

It has been moved to Figure EV4F and 4G.

4- In figure 7 MCIDAS and E2F4 are overexpressed together, in my opinion there is a lack of controls (overexpression of the 2 alone), at least the WB in figure 7a should be repeated with the single overexpression.

We used this construct to overexpress both MCIDAS and E2F4, as it has been extensively used to induce multiciliation in fibroblasts. The overexpression of the single cDNA fails to induce multiciliation. The primary purpose of this experiment was to check whether rapamycin would affect the trigger of multiciliation. This was not the case, indicating that mTORC1 acts upstream of these multiciliation factors. The observation that multiciliation increases mTORC1 activity was surprising and we believe it is interesting to report, even without dissecting each agent alone.

5- In the same figure the Authors claim that overexpressing MCIDAS and E2FA increases mTORC1 activity, quantification in 7b shows that only pS6 is increased statistically, whereas pS6K and pCAD are not.

This claim of mTORC1 increased activation is fascinating but should be strengthened for example by checking also p4EBP1 or in alternative since it is not a key part of the actual story could be reconsidered for future work, it feels also a bit too speculative in the

discussion. The single overexpression of MCIDAS and E2F4 could also help clarify if either of them can activate mTORC1 on its own.

We run additional samples and provide further quantification. As shown in the new figure 6A, in each of the three independent experiments there is an up-regulation of canonical mTORC1 signaling (p-CAD and p-S6) and a concomitant down-regulation of mTORC2 signaling (p-Akt and p-GSK3).

Very minor comments:

1- there is a reference of figure 5D which is not present, I imagine the authors meant 4D.

The reference to Fig 4D has been corrected.

2- In the phosphoproteomic screening chapter Authors wrote: out of 57 mTORC1 substrates reported in a recent review etc, without citing the review, this should be added.

The citation has been added.

Referee #3:

In this manuscript, Bakolé, Srivastava and colleagues perform an extensive body of work to uncover new roles of mTOR signalling across ependymal cell differentiation. Using gain and loss of function studies in vivo and in vitro and a range of technological approaches including metabolic profiling and phosphoproteomics, the authors demonstrate that mTORC1 is necessary for cell growth, metabolic rewiring, centriole amplification and disengagement during ependymal cell differentiation. The authors go on to show that inhibition of mTORC1 blocks the expression of cell cycle genes involved in the multiciliation cell cycle and centriole amplification, blocking ependymal cell differentiation. mTORC1 acts upstream of Mcidas, a master regulator of ependymal cell differentiation, and Mcidas can, in turn, activate mTORC1 during ependymal cell differentiation. Using phosphoproteomic and photo-mutant analysis, the authors discover new mTORC1 targets during ependymal cell differentiation and show that mTORC1-mediated phosphorylation of Gas2L1 drives centriole disengagement.

The work is convincing and the phenotypes presented are strong. The study is also timely as it builds upon recent discoveries, including the role of mTOR in regulating the size of cells lining the brain lateral ventricles and the multiciliation cell cycle, and will be

of great interest to the cell and molecular biology communities as well as developmental biologists interested in brain development and developmental disorders such as hydrocephalus.

There are some minor issues that the authors should address before publication in EMBO reports.

1. The manuscript is well written, but the amount of information and 9 figures makes it challenging to read. To highlight the main points of the story, I recommend shortening the text and reducing the number of figures by 1-2. For example, the data presented in Figure 1 is important, but it is not novel so it could be made a supplementary figure. Figure 5 does not add to the main points of the story so it could also be made a supplementary figure and the text related to it shortened. Adding a summary figure will be great to bring together the many and important findings of the study.

We reduced the number of figures to 8. We moved Figure 5 to Figure EV4F and 4G. We added a summary figure in Panel 8G.

2. Figure 1D-F. Based on what is plotted, each dot in Figure 1D and F probably represents one mouse (one wall of the lateral ventricle). Does each dot in Figure 1F represent one cell? Please add this information in the figure legend.

Each dot corresponds to one lateral ventricle (1D and $n > 9$, E and $n > 4$) and one cell (1F and $n > 3$). The information has been added to the figure legend.

3. Figure 1G. 26% and 67% of the cells lining the lateral walls are classified as ependymal cells in the P7-P10 and P15-P45 groups, respectively. However, most ependymal cells in the brain are thought to complete their differentiation into multiciliated cells during the first postnatal week (Spassky et al. 2015). Can the authors comment on this? Is it possible that adult neural stem cells are classified as progenitor cells in this analysis? I guess one way of answering this is, does the number of FOXP1-positive cells in the lateral wall/regions analysed differ between control and rapamycin-treated pups as the authors later find in vitro?

In the analysis shown in Figure 1G, cells were categorized based on the number of mature centrioles ($cen2^+deup^-$), as depicted in Figure 1C. Consequently, adult stem cells in contact with the ventricle are categorized as "progenitor cells."

4. Figure 2. What do dots represent in each plot? How many samples were analysed for

each experiment and experimental group? Please include this information in the figure legend.

Each dot corresponds to one cell (n>3). The information has been added to the figure legend.

5. Figure legend for Figure 2. Please correct a typo in the scale bar units.

The typo has been corrected.

6. In page 10, the authors take advantage of in vitro ependymal cell cultures to study how mTORC1 regulates ependymal cell differentiation. Among others, they find that inhibiting mTOR signalling by rapamycin results in fewer FOXJ1-positive cells but activation of mTOR signalling in Tsc1 cKO cells accelerates centriole amplification but does not result in more multiciliated (potentially FOXJ1-positive) cells. From this, the authors conclude that "(...) mTORC1 activation triggers ependymal cell differentiation but not its specification". Because there are fewer FOXJ1-positive cells in rapamycin-treated cultures than in controls, would a more accurate conclusion be that mTOR signalling is necessary but not sufficient for ependymal cell specification?

We agree with this conclusion and modified the text accordingly.

7. To link mechanical forces (related to brain growth) and mTOR signalling, an interesting experiment would be to wash out rapamycin from cultures after 4 days and allow the cells to recover. Does the number of multiciliated cells recover as in in vivo experiments? This experiment is not required for publication, but will be interesting to know the answer.

We performed the experiment suggested by the reviewer and present the data in the figure below. In brief, primary cells were treated with DMSO (control), rapamycin at 20 nM for 8 days (Dif 1 to 8), or rapamycin at 20 nM for 4 days, followed by washout until Dif 8. At Dif 8, the cells were stained for p-rpS6, FOP, and GT335 to examine their recovery potential. As *in vivo*, cells activate mTORC1 pathway immediately after washout and ependymal differentiation occurs in these cells.

8. In page 12 "mostly due to reduced N-carbamoyl-aspartate levels (Fig. 5D)" should be Fig. 4D.

This mistake has been corrected

9. In page 16, "Out of 57 mTORC1 substrates reported in a recent review in different cell types, 14 were significantly down-regulated in rapamycin-treated ependymal cells, highlighting the ability to find target phosphorylation sites of interest using this methodology." Please add reference.

The citation has been added

10. Figure 1: include scale bars in all microscopy images.

Scale bars are included in Figure 1

11. Mouse proteins (and antibodies) should be all capitals, please revise this throughout the text and figures.

We revised this point in the text

12. The referencing style may need amending to match EMBO reports' style. Currently, the first two authors are listed in in-text references, whether they are co-first authors or not (only first or co-first authors should appear in in-text references).

We corrected this point

Dear Dr. Spassky,

Thank you for submitting your revised manuscript. It has now been seen by two of the original referees.

As you can see, referees find that the study is significantly improved during revision and recommend publication. However, I need you to address the points below before I can accept the manuscript.

- Please rename the 'Competing Interests Statement' section as 'Disclosure And Competing Interests Statement'.
- Please remove the 'Author Contributions Statements' section from the manuscript text.
- References should be moved before the figure legends. As per our format requirements, in the reference list, citations should be listed in alphabetical order and then chronologically, with the authors' surnames and initials inverted; where there are more than 10 authors on a paper, 10 will be listed, followed by 'et al.'. Please see <https://www.embopress.org/page/journal/14693178/authorguide#referencesformat>
- Please enter the complete funding information into the manuscript tracking system as well.
- We note that currently the EV figures are combined in a single PDF file. They need to be uploaded as separate files and their nomenclature needs to be updated to Figure EV1, etc. instead of Figure Expanded View EV1, etc.
- We note that there are two Figure 6 and two Figure 8 so Figure 5 and Figure 7 appear to be missing.
- We note that there are 4 Excel tables uploaded with titles Tables S1-S4. These are datasets and need to be renamed to Dataset EV1-EV4 (source file names, titles in the manuscript tracking system, manuscript callouts need to be updated accordingly).
- We note that some time lapse movies are referred to in the manuscript text, but no movies were submitted.
- We note that the source data checklist has not been filled in. I attached the empty checklist for you to fill in and return.
- We note that the source data for Figure 4 is currently missing. If this is submitted in the form of a Dataset, it should be indicated accordingly in the source data checklist.
- In the Data Availability section, please provide separate links that directly resolve to each of the datasets themselves, rather than to the main page of the database. Also, please make all datasets publicly available.
- Please cite Boudjema et al, 2024 in the following preprint format:
 - o In-text citation: (preprint: NAME1 et al, YEAR)
 - o Author NAME1, Author NAME2, (YEAR) article title. bioRxiv doi: 1234/002.dj123 [PREPRINT]
- Materials and Methods section should be renamed as Methods.
- During our routine figure checks, we noted that control image was reused between Figure EV3 C and Figure EV4 D, which is only allowed in case the experiments were carried out in parallel. In which case, please clarify this in both of the affected legends.
- Our production/data editors have asked you to clarify several points in the figure legends - Figure Legends (main + EV):
 - o Please note that the figure 5 is mislabeled as figure 6 in the manuscript. This needs to be rectified.
 - o Please note that the figure 7 is mislabeled as figure 8 in the manuscript. This needs to be rectified.
 - o Please define the annotated p values ****/**/**/* as well as provide the exact p-values for the same in the legend of figure 8C, EV3 D as appropriate.
 - o Please indicate what */**/**/* represents; if this represents p value(s), please indicate the statistical test used and where appropriate, specify the exact p value in the legend(s) of figure(s) 4D, EV5 B, C, D, F
 - o Please note that the exact p values are not provided in the legends of figures 1D-F; 2B, C, E, F, H, J, K; 3B, D, F, G; 5B, 6E, 7E, 8C, F; EV1 D, EV2 B, D, E; EV4 B, E, G.
 - o Please indicate the statistical test used for data analysis in the legends of figures 4C, 7C
 - o Please note that information related to n is missing in the legends of figures 4C, 6B, C; 7C.
 - o Please note that the error bars are not defined in the legends of figures 6B, C.
 - o Please note that the scale bar needs to be defined for figures EV1 A-C; EV2 A, C; EV3 B
 - o Please note that scale bar and its definition are missing for figure EV3 A.
- Papers published in EMBO Reports include a 'synopsis' and 'bullet points' to further enhance discoverability. Both are displayed on the html version of the paper and are freely accessible to all readers. The synopsis includes a short standfirst summarizing the study in 1 or 2 sentences (max 35 words) that summarize the paper and are provided by the authors and streamlined by the handling editor. I would therefore ask you to include your synopsis blurb and 3-5 bullet points listing the key experimental findings.
- In addition, please provide an image for the synopsis. This image should provide a rapid overview of the question addressed in the study but still needs to be kept fairly modest since the image size cannot exceed 550 (width) x 300-600 (height) pixels.

Thank you again for giving us to consider your manuscript for EMBO Reports, I look forward to your minor revision.

Kind regards,

Deniz Senyilmaz Tiebe

--

Deniz Senyilmaz Tiebe, PhD
Senior Scientific Editor
EMBO Reports

Referee #2:

The Authors have responded to my comments successfully, they performed the experiments I requested. Therefore I do not have additional comments or requests.

Referee #3:

The authors have done a great job at addressing all reviewer comments and in my view, the manuscript is now suitable for publication in EMBO reports.

All editorial and formatting issues were resolved by the authors.

Dr. Nathalie Spassky
IBENS
France

Dear Dr. Spassky,

Thank you for submitting your revised manuscript. I have now looked at everything and all is fine. Therefore, I am very pleased to accept your manuscript for publication in EMBO Reports.

Congratulations on a nice work!

Kind regards,

Deniz Senyilmaz Tiebe

--

Deniz Senyilmaz Tiebe, PhD
Senior Scientific Editor
EMBO Reports

--
